# Perceived community alignment increases information sharing

Elisa C. Baek [1,2] ✉, Ryan Hyon [2], Karina López [2], Mason A. Porter [3,4,5] & Carolyn Parkinson [2,6] ✉

It has been proposed that information sharing, which is a ubiquitous and consequential behavior, plays a critical role in cultivating and maintaining a sense of shared reality. Across three studies, we test this theory by investigating whether or not people are especially likely to share information that they believe will be interpreted similarly by others in their social circles. Using neuroimaging data collected while people who live in the same residential community viewed brief film clips, we find that more similar neural responses across participants is associated with a greater likelihood to share content. We then test this relationship using two behavioral studies and find (1) that people are particularly likely to share content that they believe others in their social circles will interpret similarly and (2) that perceived similarity with others leads to increased sharing likelihood. In concert, our findings support the idea that people are driven to share information to create and reinforce shared understanding, which is critical to social connection.

Information sharing is a ubiquitous human behavior. Interpersonal sharing of information, which can spread particularly effectively in online media, can powerfully shape people's opinions, behaviors, and attitudes across domains (ranging from health behaviors[1] to political action[2]). Additionally, it has been hypothesized that information sharing supports fundamental human motivations to connect and belong socially[3,4] and that it plays an important role in constructing and reinforcing a sense of generalized shared reality (i.e., the sense of "being on the same page"), which is critical for social connection[5,6].

Corroborating the above two hypotheses, empirical evidence suggests that anticipation of positive social interactions is a key motivation for sharing information[7,8], and recent neuroimaging work has demonstrated that activity in regions of the brain that are involved in mentalizing (i.e., understanding the mental states of others) plays an important role in information sharing. For example, regions of the brain that are associated with mentalizing (e.g., the medial prefrontal cortex, precuneus, temporal junction, and superior temporal sulcus[9,10]) are activated when people think about sharing content with

others[11]. Accordingly, when people make sharing decisions, they may spontaneously consider how others would respond to the shared information. The extent to which a piece of content engages these brain regions is associated both with neuroimaging participants' self-reported likelihoods of sharing[11] and with population-level virality (i.e., how often the content is actually shared in the real world)[12]. Behavioral evidence also suggests that the relationship between mentalizing and sharing likelihood is causal; thinking about other people's mental states and perspectives when considering content to share increases the likelihood of sharing content[13]. Collectively, these results suggest that people actively consider the mental states of other people when considering content to share and are motivated to share information to fulfill their needs to connect socially with others.

Given that having shared understanding with others is linked to social connection[14,15] and that the desire to connect socially is a key motivation for sharing behavior[7,8], one possibility is that people consider the extent to which content will cultivate shared understanding with others when deciding whether or not to share it. Therefore, the involvement of mentalizing processes in information sharing may, in

[1]Department of Psychology, University of Southern California, Los Angeles, CA, USA. [2]Department of Psychology, University of California, Los Angeles, Los Angeles, CA, USA. [3]Department of Mathematics, University of California, Los Angeles, Los Angeles, CA, USA. [4]Department of Sociology, University of California, Los Angeles, Los Angeles, CA, USA. [5]Sante Fe Institute, Santa Fe, NM, USA. [6]Brain Research Institute, University of California, Los Angeles, Los Angeles, CA, USA. ✉e-mail: elisa.baek@usc.edu; cparkinson@ucla.edu

part, reflect individuals considering the perspectives of potential information receivers to determine whether or not others would respond to the shared information in ways that evoke shared understanding. For instance, people may share information that they believe others will interpret similarly because doing so reinforces perspectives, attitudes, and beliefs about the world that are already well-established and agreed upon in their social circles; shared understanding across these various facets is important to social connection[4,14,15].

In the present paper, we investigate the idea that motivations to achieve and maintain shared reality with others may play a critical role in information sharing. We thereby provide empirical evidence that advances existing theories about the motivations behind information sharing, which have often focused on non-social drivers of sharing (e.g., the desire to spread information that fulfills a need for accuracy[16–20]). Across three studies, we test the hypothesis that people are more likely to share information when they believe that others in their social circles will share their viewpoints about the information than when they believe that others' viewpoints will differ from theirs.

In Study 1, we used functional magnetic resonance imaging (fMRI) to test whether or not people are more likely to share content when it evokes similar neural responses in members of their social circles. We used inter-subject correlations (ISCs) of neural responses while participants watched dynamic, naturalistic stimuli (i.e., videos) to capture the similarity of brain responses across participants as these responses unfold over time. Prior research has linked ISCs of neural responses to naturalistic stimuli with participants' interpretations and understanding of messages[21–23], suggesting that this approach can meaningfully capture similarities in relevant high-level psychological processes (e.g., inferring others' mental states or integrating incoming information into existing knowledge) across individuals.

The results of Study 1 support our hypothesis. We found that coordinated neural responses in brain regions that previously have been implicated in shared high-level interpretations and low-level sensory processing are associated with an increased sharing likelihood, suggesting that similarities in interpretations and understanding of messages across individuals are associated with the likelihood of sharing the messages. Building on the results of Study 1, we directly tested these associations by examining whether or not individuals are more likely to share content when they believe that others in their social circles will interpret the content similarly to themselves. Accordingly, we conducted an online behavioral study (Study 2) and found that participants were especially likely to share content when they believed that other people in their social circles would have similar views about the content as themselves. These results held even when controlling for participants' levels of interest in the content and for their evaluations of it. We then conducted an experimental study (Study 3) to test whether or not a general sense of similarity with others causally increases the sharing likelihood. The results of Study 3 suggest that this relationship is causal, with perceived alignment between individuals' broad attitudes and preferences and those of others in their social circles causally increasing sharing likelihood.

Taken together, the findings of our three studies suggest that people are more likely to share information when they believe that others in their social circles share their own viewpoints and opinions.

## Results

In Study 1, participants underwent fMRI scanning while they watched a set of video clips on a variety of topics. For details, see the Methods section and Supplementary Table 1. All participants were living in one of two social communities of a first-year dormitory in a large public university in the United States. This allowed us to test whether or not people are especially likely to share content that members of their own community interpret similarly, as indicated by their neural responses.

In each brain region (see the Methods section for details about the parcellation and the preprocessing of the fMRI data), we computed the Pearson correlation between the time series of neural responses for each pair of participants (i.e., each dyad) for each video. This yields one correlation coefficient for each unique combination of dyad, video, and brain region. See the Methods section for more details.

After the fMRI portion of Study 1, participants indicated their likelihood of sharing each video on social media on a 1–5 Likert scale (with "1 = very unlikely" and "5 = very likely"). In our primary analyses, we binarized the sharing-likelihood ratings (see the Methods section). This choice is consistent with recent studies that link neural similarity with behavioral measures[24–26]. To relate the participant-level sharing-likelihood ratings with the dyad-level neural-similarity measure, for each video, we transformed the participant-level binarized sharing-likelihood measure into a dyad-level sharing-likelihood measure. For each video, we categorized a dyad's sharing-likelihood rating as (1) {high sharing, high sharing} if both participants in the dyad had a high likelihood of sharing the video, (2) {low sharing, low sharing} if both participants in the dyad had a low likelihood of sharing the video, and (3) {low sharing, high sharing} if one participant had a high likelihood of sharing the video and the other had a low likelihood of sharing it. Unlike existing studies, which have investigated whether or not similarities in a participant-level attribute (e.g., number of friends[21] or loneliness[27]) are linked with greater neural similarity, we are interested in whether or not people are more likely to share content when it evokes similar neural responses in individuals in their social circles. Therefore, our contrasts are at the level of video–dyad combinations.

For each brain region, we fit a linear mixed-effects model with crossed random effects to account for the dependence structure of the data[28] (see the Methods section). In each model, the ISC in the brain region is the dependent variable, the dyad-level sharing-likelihood measure is the independent variable, and similarities in participants' age, gender, and country of origin are the control variables.

We then conducted a planned-contrast analysis[29] to identify brain regions for which a high sharing likelihood is associated with more coordinated neural responses than a low sharing likelihood (i.e., $ISC_{\{high\ sharing,\ high\ sharing\}} > ISC_{\{low\ sharing,\ low\ sharing\}}$). We focus on the contrast $ISC_{\{high\ sharing,\ high\ sharing\}} > ISC_{\{low\ sharing,\ low\ sharing\}}$, as this contrast is our most direct test of the hypothesis that people are more likely to share content that different individuals interpret similarly than to share content that different individuals do not interpret similarly. In Supplementary Fig. 1, we show our results for our exploratory contrasts $ISC_{\{high\ sharing,\ high\ sharing\}} > ISC_{\{low\ sharing,\ high\ sharing\}}$ and $ISC_{\{low\ sharing,\ high\ sharing\}} > ISC_{\{low\ sharing,\ low\ sharing\}}$. For each contrast, we performed two-tailed tests and employed Holm–Bonferroni correction to correct for multiple comparisons across brain regions. We also performed analyses to examine relationships between neural similarity and a non-binarized version of the sharing-likelihood ratings.

In Study 1, we observed that there were larger ISCs in the temporoparietal junction, superior parietal cortex, and regions of the visual cortex when participants were very likely to share information (i.e., {high sharing, high sharing}) than when participants were unlikely to share information (i.e., {low sharing, low sharing}) (see Fig. 1c). In Supplementary Table 2, we give our complete set of results for subcortical brain areas. We obtained similar patterns in the results of our exploratory contrasts (i.e., $ISC_{\{high\ sharing,\ high\ sharing\}} > ISC_{\{low\ sharing,\ high\ sharing\}}$ and $ISC_{\{low\ sharing,\ high\ sharing\}} > ISC_{\{low\ sharing,\ low\ sharing\}}$; see Supplementary Fig. 1), our analyses with a non-binarized version of the sharing-likelihood variable (see Supplementary Fig. 2), and our analyses using an alternative statistical-modeling approach (see Supplementary Fig. 3).

The results of Study 1 demonstrate that similar neural responses across individuals in a social community are associated with a greater likelihood of sharing content. Combined with previous observations that decisions to share content involve the brain's mentalizing

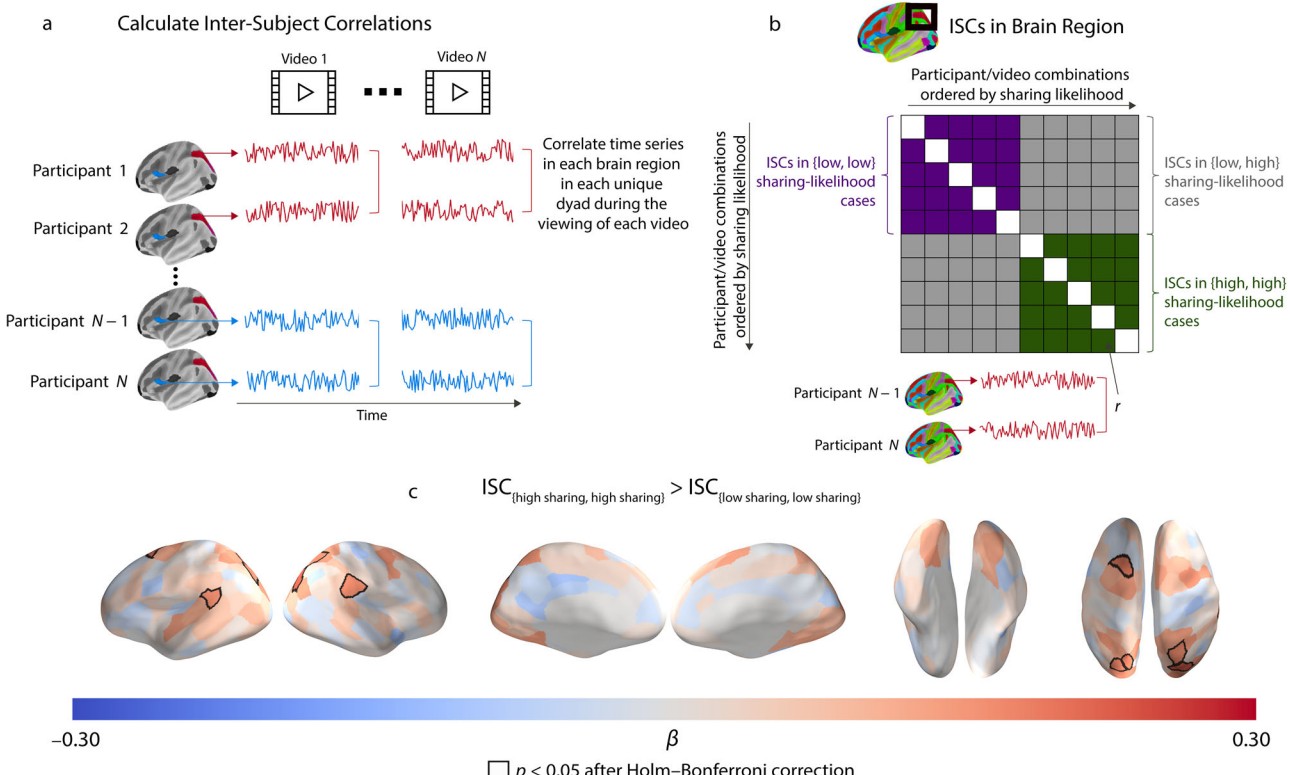

**Fig. 1 | Similar neural responses in members of a social community are associated with increased likelihood of information sharing. a** We extracted time series of neural responses while participants watched each video. For each unique dyad (i.e., pair of participants), we calculated inter-subject correlations (ISCs) of these time series for each of the 214 brain regions for each video. **b** We related neural similarities with participants' self-reported likelihoods of sharing the videos. Each cell of the matrix consists of the ISC between two participants for a brain region. The rows and columns of the matrix are ordered by participants' sharing-likelihood ratings. We performed planned contrasts of the different sharing-likelihood ratings to test whether or not there was a larger ISC when both individuals in a dyad indicated a high sharing likelihood (i.e., $ISC_{\{high\ sharing,\ high\ sharing\}}$) than when both individuals in a dyad indicated a low sharing likelihood (i.e., $ISC_{\{low\ sharing,\ low\ sharing\}}$). **c** There were larger ISCs in the temporoparietal junction, superior parietal cortex, and regions of the visual cortex when participants were very likely to share than when participants were unlikely to share. The quantity β is the standardized regression coefficient that we obtain from linear mixed-effects models. We used two-tailed statistical tests and employed Holm–Bonferroni correction to correct for multiple comparisons across brain regions. [The figures in **a** and **b** are adapted from prior work: Baek et al. In-degree centrality in a social network is linked to coordinated neural activity. *Nat. Commun.* 13, 1118 (2022) and Baek, E. C., Hyon, R., López, K., Porter, M. A. & Parkinson, C. Lonely individuals process the world in idiosyncratic ways. *Psychol. Sci.* 34, 683–695 (2023). Both papers were published under the Creative Commons Attribution 4.0 International License (see https://creativecommons.org/licenses/by/4.0/). The authors created brain visualizations using FreeSurfer, which is an open-source neuroimaging toolkit for processing, analyzing, and visualizing human brain magnetic-resonance images. It is available at https://surfer.nmr.mgh.harvard.edu/fswiki/FreeSurferSoftwareLicense].

system[11,12], these results are consistent with the possibility that people may be driven to share content when they believe that others in their social circles will similarly interpret that content. Notably, the results in Study 1 have potential alternative explanations. For example, when an individual finds that particular content is engaging, there can be both less mind-wandering (and hence greater alignment with others' neural responses[30]) and a greater desire to share that content. This possibility does not require participants to be aware that the content that they rate as more worthy of sharing also elicits similar responses in others. When content is engaging, people may both have especially similar neural responses to it and be particularly likely to share it with others without necessarily realizing that the content may evoke very similar responses across perceivers. Therefore, in Study 2, we directly tested the hypothesis that people are more inclined to share content that they believe will elicit similar responses in others in their social circles through a preregistered online behavioral study of 100 participants. (See the Methods section for more information.) In this study, participants rated news articles on the extent to which they believed others in their social circles would share their views about the content (on a scale with the anchors "these people may or may not share my view" and "I am confident that most of these people would share my view"), how likely they were to share each article on social media, the extent to

which they believed that their social-media friends would find the article interesting, and the extent to which they believed that their social-media friends would find the article positive or negative (i.e., its valence). To address limitations in Study 1 from the fixed order of the stimuli and the time gaps between stimulus presentations and sharing-likelihood ratings, we randomized the order of the stimuli in Study 2. Additionally, participants answered questions about their likelihood to share each piece of content shortly after viewing the stimuli. See the Methods section for more details.

We now present the results of Study 2. To test our hypothesis that people are more likely to share content that they believe will be interpreted similarly by others in their social community, we fit a linear mixed-effects model to account for the dependence structure of the data[28] (see the Methods section) with sharing likelihood as the dependent variable and perceived-similarity rating as the independent variable. We found a positive association between perceived similarity and sharing likelihood ($t(492) = 9.657$, $p < 0.001$, two-tailed, $β = 0.398$, 95% CI [0.317, 0.479]; see Fig. 2), indicating that participants were more likely to share information when they believed that others in their social circles would share their views about the content. (Here and for all reported results, $t$ denotes the $t$-statistic, the number in parentheses indicates the number of degrees of freedom, $p$ denotes the $p$-value, β is

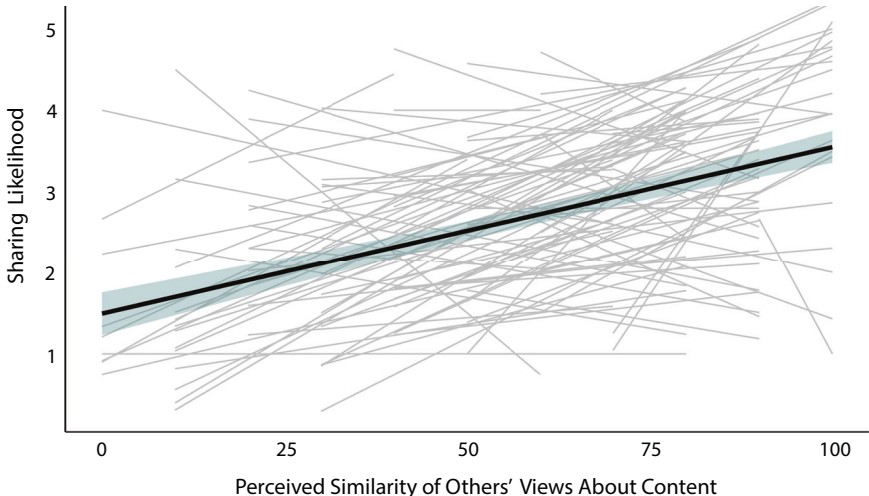

**Fig. 2 | Participants are more likely to share content when they believe that others will interpret the content similarly to themselves.** There was a positive association between perceived similarity and sharing likelihood in Study 2. That is, the study participants were more likely to share information when they believed that others in their social circles would have similar views of the content as themselves. The black line gives the mean group-regression line, the light blue band indicates the 95% confidence interval, and the light gray lines are participant-level regression lines. We measured perceived similarity on a scale with the anchors "0 = these people may or may not share my view" and "100 = I am confident that most of these people would share my view". See the Methods section for more details.

the standardized regression coefficient, and CI denotes the confidence interval.) Given prior work that suggests links between information sharing and both the valence of content and the extent to which content is perceived as interesting[17,31,32], we also fit a linear mixed-effects model with sharing likelihood as the dependent variable, perceived-similarity rating as the independent variable, and participants' interest and valence ratings as control variables. We found that the association between perceived similarity and sharing likelihood remained significant even after controlling for interest and valence ratings ($t(495) = 4.965$, $p < 0.001$, two-tailed, $\beta = 0.189$, 95% CI [0.115, 0.265]). This suggests that the link between perceived similarity and sharing likelihood does not arise merely because people are more likely to share and to have similar perceptions of information that is more interesting, extremely positive, or extremely negative. The results of Study 2 support our interpretations of our neuroimaging findings from Study 1, suggesting that people are more likely to share content that they believe will evoke similar interpretations across different individuals.

In Study 3, we conducted a behavioral experiment. Given the results of Study 2, which suggest that there is an association between the perceived similarity of others' views about a specific piece of content and their likelihood to share that content, we tested whether or not a general sense of similarity with others causally increases the sharing likelihood. Accordingly, we conducted a preregistered online experimental study to test whether or not participants are more likely to share information on social media with others who broadly hold similar views and preferences as themselves than with others who hold dissimilar views and preferences. This builds on Study 2 to test the theory that participants are more likely to share information with others who tend to have similar beliefs, preferences, and demographic characteristics as themselves because presumably such similarly-minded people are also more likely to share their views on diverse types of content. It is possible that asking participants about sharing likelihood and perceived similarity in close succession in Study 2 increased the chance that participants gave similar responses to both questions. Study 3 alleviates this concern by experimentally manipulating perceived similarity and having participants report only their sharing likelihoods.

In Study 3, 300 participants first answered a series of questions about their demographic information and their preferences about a

variety of topics (e.g., movies, news sources, and television shows). (See the Methods section for more details.) Participants were then given a choice of five news articles and selected the article in which they were most interested. They were then assigned uniformly at random to one of four experimental conditions. In each condition, participants were asked to consider sharing the news article with a Facebook group with a particular social context[33]: (1) participants in the "similar social context" condition were told that the majority of other people in the Facebook group were similar to them in demographic characteristics and preferences; (2) participants in the "dissimilar social context" condition were told that the majority of other people in the group were dissimilar to them; (3) participants in the "unclear social context" condition were told that it was not clear whether or not other people in the group shared their demographic characteristics and preferences; and (4) participants in the "mixed social context" condition were told that some people in the group were similar to them and others were different from them in their demographic characteristics and preferences. All participants were then asked to indicate their likelihood of sharing the article that they had chosen earlier with the Facebook group.

Our main hypothesis in Study 3 was that participants would be more likely to share information with others whom they believed were similar to themselves in views, preferences, and demographic characteristics (and hence presumably would respond similarly to content) than with others whom they believed were different from themselves in views, preferences, and demographic characteristics. To test this hypothesis, we first fit a linear-regression model with sharing likelihood as the dependent variable and the experimental condition (i.e., social context) as the independent variable. We then performed a planned-contrast analysis[29] to test whether or not there was a greater sharing likelihood when participants considered sharing content with others whom they believed had very similar views, preferences, and demographic characteristics to their own than when they considered sharing with others whom they believed were dissimilar to themselves (i.e., similar > dissimilar). Given that individual differences in baseline sharing (i.e., how often an individual generally shares content online) and level of interest in the content are likely to affect participants' sharing likelihoods, we also fit an additional model and performed a planned-contrast analysis with baseline sharing and interest ratings as

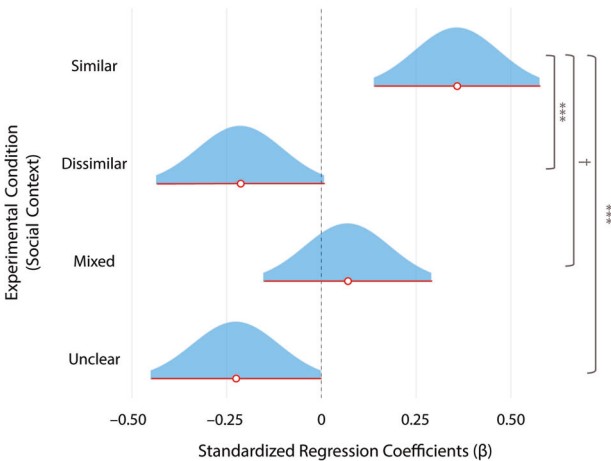

**Fig. 3 | Perceived community alignment increases participants' sharing likelihoods.** Participants in Study 3 (with *n* = 300 participants) were more likely to share information with a group when they believed that the people in that group had similar demographic characteristics, views, and preferences as themselves and thus presumably would respond similarly to content as themselves (i.e., similar > dissimilar). Participants were also more likely to share with a group of others whom they perceived as similar to themselves than with a group in which it was unclear whether or not the people in it were similar to themselves (i.e., similar > unclear). Participants were not significantly more likely to share with a group of others whom they perceived as similar to themselves than with a group in which they perceived some people as similar and others as dissimilar (i.e., similar > mixed). See Supplementary Tables 3 and 4 for the results of all examined contrasts. The white circles indicate regression coefficients from a linear model that predicts sharing-likelihood ratings from experimental condition (i.e., social context). The red lines indicate 95% confidence intervals of the coefficients, and the blue regions indicate the associated distributions. The symbol *** denotes a *p*-value of *p* < 0.001, and the symbol † denotes a *p*-value of *p* < 0.10. (The *p*-value is *p* = 0.092 for the similar > mixed contrast.) We used two-tailed statistical tests and employed false-discovery-rate (FDR) correction to correct for multiple comparisons due to multiple contrasts.

control variables. Furthermore, although the similar > dissimilar contrast is the most direct test of our main hypothesis, we explored whether or not participants would be more likely to share information with a group of similar others than with a group of others whom they believed had mixed characteristics, views, and preferences (i.e., similar > mixed) or with a group of others in which it was unclear whether or not they shared their characteristics, views, and preferences (i.e., similar > unclear). We report the results of all possible contrasts in Supplementary Tables 3 and 4. For all of our analyses, we employed false-discovery-rate (FDR) correction to correct for multiple comparisons due to multiple contrasts.

We now present the results of Study 3. As hypothesized, we found that participants were more likely to share information with others whom they perceived as similar to themselves than with others whom they perceived as dissimilar (i.e., similar > dissimilar) ($t(296) = 3.612$, $p_{corrected} = 0.001$, two-tailed, $\beta = 0.572$, 95% CI [0.151, 0.992]; see Fig. 3). The results held even when controlling for participants' baseline sharing and interest ratings ($t(294) = 4.453$, $p_{corrected} < 0.001$, two-tailed, $\beta = 0.625$, 95% CI [0.252, 0.998]). Participants were also more likely to share information with a group of others whom they perceived as similar to themselves than with a group in which it was unclear whether or not the people in it shared their views (i.e., similar > unclear) ($t(296) = 3.659$, $p_{corrected} = 0.001$, two-tailed, $\beta = 0.583$, 95% CI [0.160, 1.006]). Participants were not significantly more likely to share information with a group of others whom they perceived as similar to themselves than with a group of others that they perceived as including some people who shared their views and others who did not share them (i.e., similar > mixed) ($t(296) = 1.824$, $p_{corrected} = 0.092$, two-tailed, $\beta = 0.289$, 95% CI [−0.132, 0.709]).

## Discussion

What drives information sharing? Across three studies, we found that people are more likely to share information when they believe that it will be interpreted similarly by others in their social community. We found that inter-subject neural similarities in several regions of the brain, including both regions that are associated with low-level sensory processing and regions that are associated with high-level cognitive processing, were correlated with sharing likelihood. Accordingly, our findings suggest that information is more likely to be shared when it engages individuals' brains in similar ways, capturing their attention in a coordinated fashion. In concert with prior work that highlights the involvement of the brain's mentalizing system during decisions to share information[11,12,34], our results suggest that people's decisions to share content may be driven partly by the extent to which they believe that it will be processed similarly by others in their social communities. Indeed, our behavioral studies that directly test these relationships provide evidence that perceived similarity causally increases information sharing. Specifically, we found that people were more likely to share content when they believed that others would share their viewpoints and opinions about it. Taken together, our findings are consistent with theories of information sharing as an inherently social behavior that contributes to the formation and reinforcement of shared realities, which in turn promote social connection and cohesion[3,5].

Brain areas in which coordinated activity was associated with increased sharing likelihood included regions of the temporoparietal junction that are part of the default mode network. These regions have been implicated previously in social cognitive processes such as mentalizing (e.g., taking the perspective of others)[9,10,35], and the magnitude of brain activity in these regions has been linked to both individual and population-level sharing behavior of short text-based content[11,12]. Our work expands on these findings to show that the extent to which complex, dynamic messages evoke greater coordinated activity in these regions is linked to the likelihood that content is shared. Furthermore, as inter-subject similarity of neural responses in regions of the default mode network has been associated with shared interpretations and understanding of narratives[21,22], one interpretation of our results is that people are more likely to share content that evokes a sense of collective meaning in their social environment. Accordingly, our results align with prior work that found that people are more likely to share content that they believe will strengthen their social relationships[17,36]. Our work also suggests that one way that people strengthen relationships is by sharing content that reinforces agreed-upon attitudes and beliefs.

We also found an association between greater sharing likelihood and inter-subject similarity of neural responses in brain regions that are associated with attention allocation (e.g., the superior parietal lobule) and low-level sensory cortices (e.g., regions of the visual cortex). One possibility is that messages that people feel are worthy of sharing capture and coordinate individuals' attentional processes. Indeed, there is evidence that neural responses in the dorsal attention network and low-level sensory cortices not only align when people are exposed to the same naturalistic stimuli[37,38], but also coordinate across individuals to the extent that they exhibit similar higher-level processing of the stimuli[39,40]. Accordingly, our findings that implicate similarities in the brain's higher-level cortical systems, such as regions that are involved in attention allocation and regions of the default mode network, in increased sharing likelihood suggest that this alignment of the low-level sensory regions may be due to top-down modulations that are driven by attentional and social motivations[41–44].

The results of Study 1 suggest that similar neural responses across individuals in a social community is associated with a greater sharing likelihood. In conjunction with prior observations that decisions to share information involve the brain's mentalizing system[11,12,34], these results suggest that people may be driven to share content when they believe that others in their social circles will interpret and respond to

the content similarly to themselves. In two preregistered follow-up studies (Studies 2 and 3), we directly tested whether or not individuals are more likely to share content when they believe that others in their social circles will interpret the content similarly to themselves. We thereby directly tested our hypothesis against potential alternative explanations of the neural results (for instance, that content that is more vivid or exciting may entrain brain activity and also be more likely to be shared, regardless of whether or not participants believed that others would view the content similarly). In Study 2, we found that people were more likely to share content when they believed that others in their social circles would have similar views as themselves about it. In Study 3, we found evidence that a general sense of similarity with others causally increases sharing likelihood, presumably in part because such similarly-minded others may also be more inclined to share their viewpoints on a variety of topics. Specifically, we found that people were more likely to share content when they perceived that potential receivers of that content held similar views as themselves than when they perceived that the potential receivers held dissimilar or unclear views. Accordingly, the results of our three studies corroborate theories of information sharing as an inherently social behavior[5,13,45] that supports fundamental human motivations to connect and belong[3], rather than theories that emphasize non-social motivations (such as a desire for accuracy)[16–20]. Given that shared understanding is important to social connection[14,15,26], our findings suggest that, by sharing information, individuals create and establish collective meaning that promotes social connection through shared worldviews with others around them.

The stimuli in our studies included a variety of different topics and themes (e.g., a scientific demonstration, comedy clips, and social issues in Study 1; see Supplementary Table 1). Therefore, we are unable to make strong claims about specific message-level characteristics that may influence the effects that we found between perceived similarity and sharing likelihood. However, our results illustrate that content—regardless of its specific theme or domain—is more likely to be shared when individuals expect others to interpret the content similarly to themselves. We see this in the coordinated neural responses in Study 1, the self-report data in Study 2, and the experimental manipulation in Study 3. Our findings highlight fundamental neurobiological and psychological processes that motivate and predict sharing behavior across different content characteristics. Future work that explores these effects for different types of content (e.g., political content, morally-charged content, controversial content, and others) can test whether the relationship between perceived similarity and sharing likelihood is affected by the content type (e.g., if these effects are heightened or reduced in certain contexts).

In Study 1, the videos were not presented in isolation; instead, they were presented in a fixed order amidst a stream of other content. Therefore, comparisons of the 14 videos in Study 1 may have influenced participants' likelihood to share. Although this setting has analogues in daily life experiences, where individuals watch videos on a variety of social media (e.g., TikTok, YouTube, and Instagram) in sequences that are affected by platforms' algorithms and still compare pieces of content to one another when determining shareworthiness, future work can help elucidate the effects of contextual factors (such as the order in which stimuli are presented) on sharing likelihood.

It also remains unclear whether our results still apply in contexts in which individuals have overt motivations to seek different viewpoints from their own when sharing content (e.g., when one seeks critiques of content or is unsure of how to interpret content). In such contexts, perceived similarity in viewpoints with others may not be a key driver of information sharing. Furthermore, Study 1 participants were young adults, and Studies 2 and 3 used online convenience samples in the United States. Future work can clarify whether our findings generalize across diverse contexts and populations. Moreover, future studies that include forms of information sharing that do

not involve social media may provide further insight into whether our findings also hold for other sharing contexts (e.g., offline sharing of information by people who are not regular users of social media).

Our findings also have potential applications to studying various consequential phenomena in information sharing. For instance, one can use the links between perceived similarity and sharing likelihood as a theoretical framework to study the motivations that lead to the spread of misinformation, which has widespread negative consequences[46,47]. One potential future direction is testing whether individuals' proclivity to share information when it evokes similar responses across members of their social circles may cause them to be insufficiently concerned about the accuracy of content before sharing it. One can also use a theoretical framework that is based on our results to improve the design of messaging campaigns. For instance, public-service announcements that are more likely to be interpreted similarly across individuals in a social community may be more likely to lead to message-congruent behavior, which can have a positive impact for pro-social and pro-health messages. Indeed, in one study, media content that elicited more similar neural responses across individuals in a small group of participants also elicited higher levels of real-world engagement beyond that sample[48]. Additionally, effective speeches elicit more similar neural responses across individuals than ineffective speeches[49]. It seems particularly fruitful for future work to explicitly test whether or not similarly-interpreted content is more effective and more likely to be shared.

In summary, our results suggest that individuals are more likely to share information when they believe that it will be interpreted similarly by others in their social circles. We found that coordinated neural responses across individuals were associated with increased sharing. In subsequent behavioral studies, we found convergent evidence that individuals were more likely to share content when they believed that others in their social circles would hold similar viewpoints as themselves. In concert, our findings support the idea that information sharing plays a critical role in creating and reinforcing individuals' shared realities, which is important to social connection.

## Methods
All procedures were carried out in accordance with ethical standards and were approved or certified as exempt by the Institutional Review Board (IRB) of the University of California, Los Angeles (UCLA). In Study 1, all participants provided informed consent in accordance with the procedures of UCLA's IRB. Studies 2 and 3 were certified as exempt by UCLA's IRB. Study 2 and Study 3 participants saw an information sheet in accordance with the procedures of UCLA's IRB.

### Study 1: fMRI study
**Study participants.** A total of 70 participants participated in our fMRI study. All participants were living in one of two communities of a first-year dormitory in a large public university in the United States. Participants received $50 for their participation in the study. We tested whether or not participants were more likely to share content that they felt would be interpreted similarly, as indicated by similar neural responses, by others in their social community. We excluded all data from four participants. One participant did not complete the scan, two participants had excessive head movement, and one participant fell asleep in the scan. Additionally, we included only partial data from two participants. One participant had excessive head movement in one of the runs, and one participant reported falling asleep in one of the runs. Therefore, of the 66 individuals in our analysis, we used full data from 64 of them and partial data from 2 of them. We reported on separate analyses of the Study 1 data set in manuscripts that examined other (and very different) research questions[26,27,50].

**fMRI procedure.** Participants attended a study appointment that included a 90-minute session in which they were scanned using blood-oxygen-level-dependent (BOLD) fMRI and completed a series of self-

report surveys. Prior to the fMRI portion of the study, participants completed a demographic survey, from which we obtained their self-reported ages and genders. We then informed participants that they would be watching a series of video clips in the fMRI scanner while their brain activity was measured. We also informed them that their experience would be akin to watching television while another person "channel-surfed". (The term "channel-surfing" is an idiom that refers to scanning through different television channels.) We instructed the participants to watch the videos naturally, as they would in real life. In the scanner, participants watched 14 video clips with sound that ranged in duration (from 91 to 734 seconds) and content. (See Supplementary Table 1 for descriptions of the content.) The video task was divided into four runs, and the total task lasted approximately 60 minutes. All participants saw the videos in the same order. (We performed a permutation test and found that there was no statistically significant relationship between sharing likelihood and when a video clip appeared in the stimulus sequence. See the Supplementary Information for more information.) After the fMRI scan, participants indicated their likelihood to share each video on social media by answering the question "How likely would you be to share this video on social media?" with the anchors "1 = very unlikely" and "5 = very likely" (as used in prior work[11]).

**fMRI data acquisition.** We acquired neuroimaging data using a 3T Siemens Prisma scanner with a 32-channel coil. The functional images were recorded using echo-planar sequences (with echo time = 37 ms, repetition time (TR) = 800 ms, slice thickness = 2.0 mm, voxel size = 2.0 mm × 2.0 mm × 2.0 mm, matrix size = 104 × 104 mm, field of view = 208 mm, multi-band acceleration factor = 8, and 72 interleaved slices with no gap between them). To allow stabilization of the BOLD signal, we added a "start" buffer (with a duration of 8 seconds) and an "end" buffer (of 20 seconds) to the beginning and end of each run, respectively. Participants saw a blank black screen during these buffers. We also acquired high-resolution T1-weighted (T1w) images (with echo time = 2.48 ms, repetition time = 1900 ms, slice thickness = 1.0 mm, voxel size = 1.0 mm × 1.0 mm × 1.0 mm, matrix size = 256 × 256 mm, field of view = 256 mm, and 208 interleaved slices with a 0.5 mm gap between them) to use in coregistration and normalization. To minimize head motion, we attached adhesive tape to the headcase and stretched it across participants' foreheads[51].

**fMRI data analysis.** We used fMRIPrep version 1.4.0 for the data processing of our fMRI data[52]. We have taken the descriptions of anatomical and functional data preprocessing that begins in the next paragraph from the recommended boilerplate text that is generated by fMRIPrep and released under a CC0 license, with the intention that researchers reuse the text to facilitate clear and consistent descriptions of preprocessing steps, thereby enhancing the reproducibility of studies.

For each participant, the T1-weighted (T1w) image was corrected for intensity non-uniformity (INU) with N4BiasFieldCorrection, distributed with ANTs 2.1.0[53], and used as a T1w-reference throughout the workflow. Brain-tissue segmentation of cerebrospinal fluid (CSF), white matter (WM), and gray matter (GM) was performed on the brain-extracted T1w using FSL FAST[54]. Volume-based spatial normalization to the ICBM 152 Nonlinear Asymmetrical template version 2009c (MNI152NLin2009cAsym) was performed through nonlinear registration with antsRegistration (ANTs 2.1.0)[53].

For each of the four BOLD runs per participant, the following preprocessing was performed. First, a reference volume and its skull-stripped version were generated using a custom methodology of fMRIPrep. The BOLD reference was then coregistered to the T1w reference using FSL FLIRT[54] with the boundary-based registration cost function. The coregistration was configured with nine degrees of freedom to account for remaining distortions in the BOLD reference. Head-

motion parameters with respect to the BOLD reference (transformation matrices and six corresponding rotation and translation parameters) were estimated before any spatiotemporal filtering using FSL MCFLIRT[54]. Automatic removal of motion artifacts using independent component analysis (ICA–AROMA) was performed on the preprocessed MNI-space BOLD time series after removal of non-steady-state volumes and spatial smoothing with an isotropic, Gaussian kernel of 6 mm FWHM (full-width half-maximum). The BOLD time series were then resampled to the MNI152Nlin2009cAsym standard space.

The following 10 confounding variables generated by fMRIPrep were included as nuisance regressors: global signals extracted from within the cerebrospinal fluid, white matter, and whole-brain masks; framewise displacement; three translational motion parameters; and three rotational motion parameters.

**Cortical parcellation into brain regions.** We extracted neural responses across the whole brain for each video using the 200-parcel cortical parcellation scheme of Schaefer et al.[55] and 14 subcortical regions using the Harvard–Oxford subcortical atlas[56]. Together, this resulted in 214 regions that span the whole brain.

**Inter-subject correlations (ISCs).** We used the SciPy 1.5.3 library[57] in Python 3.7.0 to calculate ISCs. We extracted the mean time series in each of the 214 brain regions for each participant at each time point [i.e., at each repetition time (TR)]. Our analyses included 66 participants after the various exclusions, so there were 2145 unique dyads. For each unique combination of dyad and video, we calculated the Pearson correlation between the mean time series of the neural responses in each of the 214 brain regions. We then Fisher z-transformed the Pearson correlations and normalized the subsequent values (i.e., using z-scores) within each brain region.

**Relating neural similarity with information-sharing ratings.** As we described in the Results section, we wanted to test whether or not sharing likelihood is associated with neural similarity. To do this, we first binarized the sharing-likelihood ratings into a group with a high sharing likelihood and a group with a low sharing likelihood. The mean sharing-likelihood rating was 2.06 and the median was 2, so we classified sharing-likelihood ratings of 1 and 2 as "low likelihood" and sharing-likelihood ratings of at least 3 as "high likelihood". To relate this participant-level sharing-likelihood measure with the dyad-level neural-similarity measure, we transformed the participant-level sharing-likelihood measure for each video into a dyad-level measure. We did this by creating a binary variable that indicated whether, for each video, both participants in a dyad had a high likelihood of sharing the video ({high sharing, high sharing}), both participants had a low likelihood of sharing the video ({low sharing, low sharing}), or one participant had a low likelihood of sharing the video and the other had a high likelihood of sharing it ({low sharing, high sharing}). Of the 29,770 unique pairs of ratings, 3485 were {high sharing, high sharing}, 14,963 were {low sharing, low sharing}, and 11,193 were {low sharing, high sharing}. In Supplementary Fig. 4, we report analyses on a subset of the data that uses matching numbers of observations across the various sharing-likelihood levels.

To relate the dyad-level and video-level sharing-likelihood variables with neural similarity, we used the method in Chen et al.[28] and fit linear mixed-effects models with crossed random effects using LME4 and LMERTEST in R[58]. This approach allowed us to account for non-independence in our data from repeated observations for each participant (i.e., because each participant is part of multiple dyads), each video (i.e., because each video was rated by multiple participants), and the interaction between each participant and each video (i.e., because each participant in a dyad rated each video). Following the method that was outlined in Chen et al. (2017), we "doubled" the data (with redundancy) to allow fully-crossed random effects. In other words, we

accounted for the symmetric nature of the ISC matrix and the fact that each participant contributes twice to each data point for each dyad (because $(i, j) = (j, i)$ for participants $i$ and $j$). We then manually corrected the degrees of freedom to $N - k$, where $N$ is the number of unique observations (in our case, $N = 29,770$) and $k$ is the number of fixed effects in the model, before performing statistical inference. All findings that we report in the present paper use the corrected number of degrees of freedom.

For each of our 214 brain regions, we fit a mixed-effects model, with ISCs in the corresponding brain region as the dependent variable and the dyad-level and video-level binarized sharing-likelihood variable as the independent variables. We also included similarities in participants' age, gender, and country of origin (which we define as the country in which an individual was living prior to enrolling at the university) as control variables. We used random intercepts for each individual in a dyad (i.e., participant 1 and participant 2), each video, and the interaction between each individual and each video. To control for similarities in demographic variables, for each unique dyad (i.e., for each pair of individuals) in the fMRI session, we computed the absolute value of the difference between the ages of the two individuals in the dyad (i.e., age difference = $|age_1 - age_2|$). We then transformed this difference score into a similarity score so that larger numbers indicate greater similarity (specifically, age similarity = $1 - [$age difference/max(age difference)$]$). To control for similarities in gender, we used an indicator variable in which 0 signifies different genders and 1 signifies the same gender. To control for similarities in home country, we used an indicator variable in which 0 signifies different home countries and 1 signifies the same home country. We then included these variables (i.e., similarities in age, gender, and home country) as control variables in our models that relate ISC and sharing likelihood.

We then conducted planned contrasts using EMMEANS in R to identify the brain regions in which the ISCs were larger when participants indicated a higher likelihood to share a video than when they indicated a lower likelihood to share a video (i.e., ISC$_{\{high\ sharing,\ high\ sharing\}}$ > ISC$_{\{low\ sharing,\ low\ sharing\}}$). In Supplementary Fig. 1, we report results from the ISC$_{\{high\ sharing,\ high\ sharing\}}$ > ISC$_{\{low\ sharing,\ high\ sharing\}}$ and ISC$_{\{low\ sharing,\ high\ sharing\}}$ > ISC$_{\{low\ sharing,\ low\ sharing\}}$ contrasts. We converted all variables to $z$-scores to yield standardized regression coefficients ($\beta$) as outputs. We Holm–Bonferroni-corrected the $p$-values for multiple comparisons at $p < 0.05$. We did not formally test each model for normality of residuals or for equal variance (i.e., homoscedasticity). However, prior work has argued that our approach is robust to both error non-normality and heteroscedasticity[59]. Nevertheless, for thoroughness, we include the results of non-parametric tests that do not rely on these assumptions in Supplementary Fig. 3. These results are similar to the ones from our parametric models.

## Study 2: Correlational behavioral study
**Participants.** We recruited 100 participants who met our eligibility criteria, as outlined in the preregistration that we published on 2 January 2021 (see https://osf.io/qm4zw), on Amazon's Mechanical Turk[60]. There were no deviations from the preregistered protocol. Participants were required to have an account on social media and to report that they sometimes share content (in this case, news stories) on social media. Specifically, to be eligible to participate, participants had to answer "yes" to both of the following questions: (1) "Do you currently have an account on any of the following social media platforms: Facebook, Twitter, Instagram?"; and (2) "Do you agree with the following statement? I sometimes share news stories on social media (for example, on Facebook, Twitter, and/or Instagram)." We determined our target sample size of this online convenience sample based on power calculations using pilot data, which suggested that we would have 95% power to detect a standardized effect size of $d = 0.13$, which was the smallest estimated effect size based on pilot data.

**Procedure.** Participants completed an online survey that took 5–10 minutes and were compensated $0.85 after completing it. All participants saw the headlines and abstracts (i.e., short summaries) of five different news articles that were chosen uniformly at random from a sample of 29 news articles that we pretested in a pilot study to ensure that they (1) ranged in the extent to which their content would elicit similarity in interpretations across individuals and (2) were somewhat interesting, given that articles that are widely perceived to be uninteresting are unlikely to be shared (as a baseline), irrespective of how one believes others will interpret them.

The order of the five news articles was determined uniformly at random. Participants were asked their likelihood to share each article on social media with the question "How likely would you be to share this article on social media (e.g., on your Facebook timeline, Instagram, or Twitter)?" with the anchors "1 = extremely unlikely" and "5 = extremely likely". They also were asked the extent to which they believed that others in their social circles would have similar views as themselves about the article with the question "Consider the people with whom you are friends with on social media. How confident are you that they would all generally share your views on the content of the article?" with the anchors "0 = these people may or may not share my view" and "100 = I am confident that most of these people would share my view". To counteract potential effects of seeing one type of question before the other, participants were assigned uniformly at random to see either all of the sharing questions first (and subsequently see all of the associated perceived-similarity questions) or all of the perceived-similarity questions first (and subsequently see all of the associated sharing questions). The order of the news articles was determined uniformly at random for each set of questions. After answering all of the sharing and perceived-similarity questions, participants rated how positive or negative they thought their friends on social media would find each article and how interesting they thought their friends on social media would find each article. For the first question, they were asked "To what extent do you think your friends on social media would view the content of each article in a positive or negative light?" with the anchors "0 = extremely negative", "50 = neutral", and "100 = extremely positive". For the second question, they were asked "To what extent do you think your friends on social media would find the content of each article interesting?" with the anchors "0 = extremely uninteresting", "50 = neither interesting nor uninteresting", and "100 = extremely interesting".

**Data analysis.** To test our main hypothesis that people are more likely to share content that they believe will be interpreted similarly by others in their social community, we fit a linear mixed-effects model using LME4 and LMERTEST in R[58]. This approach allowed us to account for nonindependence in our data from repeated observations for each participant (i.e., because each participant rated multiple news articles) and each news article (i.e., because each news article was rated by multiple participants). We fit a linear mixed-effects model with sharing likelihood as the dependent variable and perceived-similarity rating as the independent variable, with random intercepts for participant and news article. We also fit a linear mixed-effects model with sharing likelihood as the dependent variable, perceived-similarity rating as the independent variable, and participants' interest and valence ratings as control variables; we again used random intercepts for participant and news article. We converted all variables to $z$-scores to yield standardized regression coefficients ($\beta$) as outputs. We did not formally test the models for normality of residuals or for equal variance. However, prior work has argued that our approach is robust to both error non-normality and heteroscedasticity[59]. Nevertheless, for thoroughness, we include the results of non-parametric tests that do not rely on these assumptions in Supplementary Fig. 5. These results suggest that our findings with parametric approaches are robust.

## Study 3: Behavioral experiment

**Participants.** We recruited 300 participants on Prolific[61] who met the eligibility criteria, as outlined in the preregistration that we published on 2 March 2022 (see https://osf.io/7tvcb). There were no deviations from the preregistered protocol. Participants from this online convenience sample were required to be regular users of Facebook. (Specifically, they needed to use it at least once a month.) We determined our target sample size based on power calculations using pilot data, which suggested that we would have 85% power to detect a standardized effect size of $d = 0.25$, which was the smallest estimated effect size based on pilot data.

**Procedure.** Participants completed an online survey that took 5–10 minutes and were compensated $0.95 after completing it. Participants first filled out their demographic information, including their age, gender, race, socioeconomic status, sexual orientation, state of residence, political ideology, and political affiliation. They then provided their preferences on various topics, including their (unordered) top-three favorite movies of all time, their favorite and least-favorite sources of news, television shows that they perceived as funny and not funny, and how they like to spend their free time. Participants were then presented with five news-article headlines and summaries, and they were asked to select the one that most interested them. As in Study 2, the five news articles were chosen to (1) range in the extent to which the content would elicit similarity in interpretations across individuals and (2) be somewhat interesting, given that articles that are widely perceived to be uninteresting are unlikely to be shared (as a baseline), irrespective of how one believes others will interpret them.

The participants were assigned uniformly at random into one of four conditions that manipulated how similar other members of a hypothetical Facebook group were to themselves: (1) similar social context, (2) dissimilar social context, (3) unclear social context, and (4) mixed social context. (See Supplementary Table 5 for the detailed instructions that the participants saw.) The participants then saw the news article that they had chosen earlier, and they were asked to indicate how likely they were to share that article with the Facebook group to which they were assigned. They were asked the question "How likely are you to share the following article with this Facebook group?" with the anchors "1 = extremely unlikely" and "5 = extremely likely". After providing their sharing-likelihood ratings, participants indicated how interesting they found the article to be and how often they typically share news articles on Facebook. For the first question, they were asked "How interesting is the following article to you?" (and they were again shown the article) with the anchors "1 = very uninteresting" and "5 = very interesting". For the second question, they were asked "How often do you share news articles on Facebook?" with the anchors "1 = less than once a year" and "5 = almost every day". We adopted our approach of experimentally assigning participants to different hypothetical Facebook groups from prior work[33].

**Data analysis.** To test our hypothesis that people are more likely to share content with others whom they perceive as similar to themselves than to others whom they perceive as dissimilar to themselves, we fit a linear-regression model in R[62]. First, we fit a linear-regression model with sharing likelihood as the dependent variable and the experimental condition (i.e., social context) as the independent variable. We then conducted a planned-contrast analysis using EMMEANS in R[63] to test whether or not participants were more likely to share content with others whom they perceived as similar than to others whom they perceived as dissimilar (i.e., similar > dissimilar). We also examined all other possible contrasts in our framework (i.e., similar > mixed, similar > unclear, mixed > dissimilar, unclear > dissimilar, and mixed > unclear). We converted all variables to $z$-scores to yield standardized regression coefficients (β) as outputs. We FDR-corrected $p$-values for multiple comparisons at $p < 0.05$. We did not formally test the model

for normality of residuals or for equal variance. However, prior work has argued that our approach is robust to both error non-normality and heteroscedasticity[59]. Nevertheless, for thoroughness, we include the results of a non-parametric test that does not rely on these assumptions in Supplementary Fig. 6. These results suggest that our findings from parametric approaches are robust.

### Reporting summary

Further information on research design is available in the Nature Portfolio Reporting Summary that is linked to this article.

## Data availability

The preprocessed data[64] for Study 1 are available at https://doi.org/10.5281/zenodo.15080346, the preprocessed data and hyperlinks to the stimuli[65] for Study 2 are available at https://doi.org/10.5281/zenodo.11106494, and the preprocessed data and hyperlinks to the stimuli[66] for Study 3 are available at https://doi.org/10.5281/zenodo.11432270. One can obtain the raw data by contacting the corresponding authors.

## Code availability

The code[64] that we used in Study 1 is available at https://doi.org/10.5281/zenodo.15080346, the code[65] that we used in Study 2 is available at https://doi.org/10.5281/zenodo.11106494, and the code[66] that we used in Study 3 is available at https://doi.org/10.5281/zenodo.11432270.

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

## Acknowledgements

We thank Meng Du, Elena Sternlicht, Kelly Xue, and the UCLA Center for Cognitive Neuroscience (particularly Jared Gilbert) for providing support with data collection. We thank Gang Chen and Mark Ho Chio Lai for their statistical advice. We thank Miriam Schwyck and Monica Thieu for their feedback on the revised manuscript. This work was supported by the National Science Foundation SBE Postdoctoral Research Fellowship (Grant no. 1911783 to E.C.B.) and the National Science Foundation (Grant no. SBE-1835239 to C.P. and M.A.P. and Grant no. SBE-2048212 to C.P.).

## Author contributions

E.C.B., R.H., M.A.P., and C.P. designed the study and experiments. E.C.B., R.H., and K.L. collected the data. E.C.B. analyzed the data (with support from C.P.). E.C.B. and C.P. wrote the original manuscript with feedback from all authors. E.C.B., C.P., and M.A.P. edited the manuscript.

## Competing interests

The authors declare no competing interests.

## Additional information

**Supplementary information** The online version contains supplementary material at https://doi.org/10.1038/s41467-025-59915-8.

**Peer-review information** *Nature Communications* thanks Mark Thornton and Yaara Yeshurun for their contribution to the peer review of this work. A peer-review file is available.

