## [Transparent Peer Review File · Nature Communications]

Perceived community alignment increases information sharing

Corresponding Author: Professor Elisa Baek

Version 0:

Reviewer comments:

Reviewer #1

(Remarks to the Author)

Baek et al. present an fMRI and behavior-based investigation of the role of perceived community alignment and information sharing. Across three studies, they find consistent evidence that people share content with others who they expect to share their views of said content. This points to a social-affiliation account of information sharing and may help shed light on problematic real world phenomena such as the sharing of misinformation.

Generally, I found this to be a very strong set of studies. The imaging study features a large sample, and a sophisticated yet appropriate methodology. The behavioral studies complement it nicely with explicit self-report and experimental manipulation. There is generally a high standard of rigor and transparency throughout this work, and it bears on an important social question. Below I make a few suggestions which I hope may improve the paper:

1. It feels as though there is a small gap in reasoning between Study 3 and the earlier studies. The earlier studies focus on how perceptions about how similarly others' would view a specific piece of content relate to sharing. In Study 3, the focus instead is not on how others' view a specific piece of content, but rather their overall similarity. In other words, although Study 3 does provide causal evidence, it seems to attest to a slightly different effect (i.e., that general similarity makes people more likely to share anything, rather than similar views about a specific piece of information making that information more likely to be shared). Although these effects support the same broad conclusion, I think the paper would benefit from acknowledging and discussing this distinction.

2. It is not clear to me that the degrees of freedom for the ISC mixed effects models have been appropriately corrected. The authors report having approximately halved the degrees of freedom from $2N - k$ to $N - k$, which is consistent with first step of the correction described in Chen et al. 2017. However, Chen et al. also describe a second step in their process: "Secondly, as there are only n independent measuring units (subjects), we discount the nominal number of DF directly from the model from $2N - k$ for each t-test." This step does not appear to have been conducted? (It is also frankly not very clearly described in the Chen et al. paper.) Moreover, in the case described, Chen et al. are discussing a case in which the only unit of observation is the participant (i.e., just 1 ISC value for each pair of participants, without the video level of observation present in the current study) which further complicates the implementation of this correction as it pertains to the current study. The end result is that I believe that the number of degrees of freedom used in the significance testing in Study 1 may have been too high. It effectively assumes that almost every pairing of videos across participants is independent, which does not seem likely to me. Given that the authors are using lmerTest anyway, I would suggest that they use the degrees of freedom estimated by the Satterthwaite approximation (i.e., the DFs reported by default in lmerTest lmer calls) instead of the Chen et al. correction, or that they determine how to fully implement the Chen et al. correction for their data structure. Alternatively, they could switch to a permutation testing based approach, as I describe in the next point below.

3. The multiple comparison approach the paper takes in Study 1 is excessively conservative. At the very least, I would suggest using the Holm-Bonferroni procedure, instead of the regular Bonferroni correction – Holm-Bonferroni has the exact same assumptions as Bonferroni, but offers slightly more statistical power. However, the ideal solution would be to use a correction method that can benefit from the dependencies between tests (i.e., dependencies between brain regions in the parcellation, and between contrasts). Using maximal statistical permutation testing would be one such approach. This could be achieved here via a modified version of the Mantel test. Importantly, this would need to be modified to appropriately deal with the structure of the data. Rather than just a single permutation of rows/columns, there would need to be a permutation

first of subjects, then of a second permutation of videos, in order to generate an appropriately structured null matrix. The mixed effects model would be fit again on each iteration of the permutation procedure, and the maximum t-statistic across regions/contrasts computed and added to the null distribution. In addition to being less conservative, an additional advantage of taking this approach is that it would obviate the need to correct degrees of freedom to obtain p-values, as discussed above.

4. The paper would be improved by a limitations/constraints on generalizability paragraph in the discussion. It feels as though there are likely to be some important contexts in which the lesson learned here may not apply. For example, if I want to hear a critique of a piece of content, I might be more likely to share it with someone who I expect to have a different viewpoint than my own than with someone who shares my opinion. There are also common limitations of sample demographics and content topics which should nonetheless be acknowledged.

Best,
Mark Thornton

Reviewer #2

(Remarks to the Author)

In this paper, across one fMRI and two behavioral studies the authors tested whether motivations to achieve and maintain shared reality with others plays a critical role in information sharing. They found that individuals are more likely to share information when they believe that it will be interpreted similarly by others in their social circles.

I think these are interesting findings on an important issue. However, I am not convinced how the fMRI study contributes to the notion that people share information with similar others and would like to better understand how the two behavioral studies add to previous research in the field. See more detailed concerns below.

Study 1 concerns:

1. The authors suggest that the results of Study 1 demonstrate that similar neural responses of individuals in a social community are associated with a greater likelihood of sharing content. They also note that the results in Study 1 have potential alternative explanations, such as participants' engagement with the videos. It makes sense to me that individuals would like to share a video they are engaged with, and previous studies suggest that when individuals are more engaged, the stimulus synchronized them more. I don't see how Study 2 rules out the role of engagement Study 1 results. I still think that Study 1 results could be explained by engagement per-se.
2. I am not sure why the contrast made in Study 1 is between high and low sharing pairs. It seems to me that in order to test their hypothesis, the contrast should be between individuals that are more or less similar to each other (regardless of sharing), which I assume will resemble the contrast done in the authors' previous paper on this dataset (Parkinson, C., Kleinbaum, A. M., & Wheatley, T. (2018). Similar neural responses predict friendship. *Nature communications*).
3. Did the authors look at differences in the association between sharing a video and neural synchronization between the different videos? I think it could be interesting to test the hypothesis that videos that elicit more controversial interpretations would show increased such association.
4. Participants provided their likelihood to share each video outside the scanner, approximately an hour after they saw the video, and after they saw all the 14 videos in the scanner. Do the authors have any way to verify that the "sharing" ratings were not affected by the comparisons between the 14 videos, and indeed reflect participant's likelihood to share the video immediately after watching it? I think this point is relevant because the order of the videos was always the same, and it could affect participant's likelihood to share the video in retrospect.
5. In their binarizing procedure, why did the authors decide to attribute "3" sharing-likelihood to "high likelihood" (Pg 24, Ln 484-5)? If I understood correctly, the scale was 1-5, thus the value "3" is relatively neutral in terms of sharing.
6. How did the authors took into account the fact that there was a (large) difference in the unique pairs of ratings? There were 3,485 {high sharing, high sharing}, 14,963 {low sharing, low sharing}, and 11,193 {low sharing, high sharing}.
7. I wonder why the authors used such a conservative threshold for multiple comparisons? Which p-value is considered significant in the correction they used (Bonferroni-corrected the p-values for multiple comparisons at $p < 0.001$)? Sometimes too conservative threshold can bias the results. For example, theoretically it could be that in some parcels the low-sharing elicited higher brain synchronization than the high-sharing, but the effect was not as significant.
8. Did the authors measure how much the participants in this study use social media? I wonder if there are people that do not use social media (or rarely use it) and this would affect their ratings of how likely they are to share the piece.

Study 2 concerns:

9. I could not locate the topics of the five articles used as stimuli. This is critical, because it is vital to understand to what extent they were controversial, and in what manner. To take an extreme example, I would imagine that it would be much easier to share a National Geographic article about Panda Bears with someone who is not similar to me (or is this perhaps also controversial these days?) than a New York Times piece about Trump's trial.
10. In Study 2, were the two questions asked immediately one after the other? ("Consider the people with whom you are friends with on social media. How confident are you that they would all generally share your views on the content of the article?" and the question "How likely would you be to share this article on social media (e.g., on your Facebook timeline, Instagram, or Twitter)?").

Moreover, why was the scale for these questions was 1-5, and not a VAS scale?

I suggest that these two characteristics of the task made it easier for the participants to make similar ratings on both questions.

11. As concern (8), did the authors measure how much the participants in this study use social media? I wonder if there are people that do not use social media (or rarely use it) and this would affect their ratings of how likely they are to share the piece.

Study 3 Concerns

12. Again, as in concern (9), I could not locate the topics of news articles. This could affect the interpretation of Study 3 results.

13. Both Study 2 and Study 3 reveal that individuals were more likely to share information when they believed that others in their social circles would hold similar viewpoints as themselves about the information. I think this is an interesting finding, that is very much in line with previous literature about the need to belong. However, I am not sure if the studies' results highlight a new perspective on it.

Version 1:

Reviewer comments:

Reviewer #1

(Remarks to the Author)

The authors have satisfactorily addressed the issues that I raised in my initial review. I particularly appreciate the lengths they went through in order to implement the different significance testing/multiple comparison approaches. Combined with the changes made to address the other reviewers' concerns, I think this already strong paper is now even stronger, and ready to make a valuable contribution to the literature.

Reviewer #2

(Remarks to the Author)

I find the revised manuscript and the analyses clearer. The authors have adequately addressed my concerns.

Response Letter

Manuscript: "Perceived community alignment increases information sharing" [NCOMMS-23-20424]

Comments from Reviewer 1

R1.1. It feels as though there is a small gap in reasoning between Study 3 and the earlier studies. The earlier studies focus on how perceptions about how similarly others' would view a specific piece of content relate to sharing. In Study 3, the focus instead is not on how others' view a specific piece of content, but rather their overall similarity. In other words, although Study 3 does provide causal evidence, it seems to attest to a slightly different effect (i.e., that general similarity makes people more likely to share anything, rather than similar views about a specific piece of information making that information more likely to be shared). Although these effects support the same broad conclusion, I think the paper would benefit from acknowledging and discussing this distinction.

Thank you for pointing out this issue. We agree that the manuscript would be strengthened by clarifying the theoretical links between Study 2 and Study 3. As Reviewer 1 notes, in Study 2, we tested the associations between individuals' sharing likelihoods and their perceived similarities with others about a specific piece of information. In Study 3, we tested how broader perceived similarities between self and others (e.g., similarities in beliefs, likes/dislikes, and demographics) lead to increased sharing likelihoods, because presumably being similar in these attributes would also imply that similar others would also share beliefs about the news articles in question.

We have made edits throughout the manuscript to (1) clarify that Study 3 tests associations between sharing likelihood and broad perceived similarities between oneself and others and (2) clarify that our overall conclusions link perceived similarities more generally with sharing likelihoods (as opposed to linking perceived similarity about a specific content with sharing likelihood).

Here are examples of some of the edits that we made (with bold typeface indicating changed text):

"We then tested this relationship using two behavioral studies and found (1) that people were particularly likely to share content that they believed others in their social circles would interpret similarly and (2) **that perceived similarity with others leads to increased sharing likelihood.**" (Abstract)

"We then conducted an experimental study (Study 3) to test whether or not **a general sense of similarity** with others causally increases the sharing likelihood. The results of Study 3 suggest that this relationship is causal, with perceived alignment between one's **broad attitudes and**

preferences and those of others in one's social circles causally increasing the sharing likelihood.” (pg. 5)

“Given the results of Study 2, which suggest that there is an association between the perceived similarity of others' views about a **specific** piece of content and their likelihood to share that content, we tested whether or not a **general** sense of similarity **with others** causally increases the sharing likelihood. Accordingly, we conducted an online experimental study (Study 3) to test whether or not participants are more likely to share information on social media with others who broadly hold similar views and preferences as themselves than with others who hold dissimilar views and preferences. **This builds on Study 2 to test the theory that participants are more likely to share information with others who tend to share similar beliefs, preferences, and traits as themselves because presumably such similarly-minded people are also more likely to share their views on diverse types of content.**” (pg. 11–12)

“In Study 3, we found evidence that a **general sense of similarity with others causally increases the sharing likelihood, presumably in part because such similarly-minded others may also be more inclined to share their viewpoints on a variety of topics.**” (pg. 18)

R1.2. It is not clear to me that the degrees of freedom for the ISC mixed effects models have been appropriately corrected. The authors report having approximately halved the degrees of freedom from $2N - k$ to $N - k$, which is consistent with first step of the correction described in Chen et al. 2017. However, Chen et al. also describe a second step in their process: “Secondly, as there are only n independent measuring units (subjects), we discount the nominal number of DF directly from the model from $2N - k$ for each t-test.” This step does not appear to have been conducted? (It is also frankly not very clearly described in the Chen et al. paper.) Moreover, in the case described, Chen et al. are discussing a case in which the only unit of observation is the participant (i.e., just 1 ISC value for each pair of participants, without the video level of observation present in the current study) which further complicates the implementation of this correction as it pertains to the current study. The end result is that I believe that the number of degrees of freedom used in the significance testing in Study 1 may have been too high. It effectively assumes that almost every pairing of videos across participants is independent, which does not seem likely to me. Given that the authors are using lmerTest anyway, I would suggest that they use the degrees of freedom estimated by the Satterthwaite approximation (i.e., the DFs reported by default in lmerTest lmer calls) instead of the Chen et al. correction, or that they determine how to fully implement the Chen et al. correction for their data structure. Alternatively, they could switch to a permutation testing based approach, as I describe in the next point below.

Thank you for this comment. We consulted Dr. Gang Chen, the first author of Chen et al., (2017)¹, for advice about our approach. He suggested that to properly account for the non-independence due to the repetition of the videos, we should expand our model to include additional terms to account for random intercepts for the interaction between each subject and video.

In this letter, we include a reference to the exact code that we used in lmerTest in R for clarity. Note that age, country, and gender are control variables. The code is as follows (with the revision in bold):

Original model (fit separately for each brain region):

ISC ~ share + age + country + gender + (1|subject1) + (1|subject2) + (1|video)

New model (fit separately for each brain region; the new terms are bolded):

ISC ~ share + age + country + gender + (1|subject1) + (1|subject2) + (1|video) +
(1|subject1:video) + (1|subject2:video)

In Figure R1, we show a side-by-side comparison of the original and new results. As the figure demonstrates, the pattern of results across the whole brain is very similar for the original and new results. Indeed, the regression coefficients between the two sets of models have a Pearson correlation of $r \approx 0.967$. The most notable difference is that significantly fewer brain parcels remain statistically significant after multiple-comparison corrections in the new model. Given the comments of both Reviewer 1 and Reviewer 2, we now use Holm–Bonferroni correction at $p < 0.05$. In our original manuscript, we included the exploratory contrasts $ISC_{\{high\ sharing, high\ sharing\}} > ISC_{\{low\ sharing, high\ sharing\}}$ and $ISC_{\{low\ sharing, high\ sharing\}} > ISC_{\{low\ sharing, low\ sharing\}}$ in our multiple-comparison corrections (i.e., for 642 tests) for the most conservative approach. However, given the feedback from both reviewers that our approach was too conservative, we now correct for multiple comparisons for the number of tests for each contrast (i.e., 214 tests). Our results with this new approach suggest that our effects may be more specific and less dispersed throughout the brain than what we previously thought.

Editorial Note: Figure R1 in this Peer Review File was created using Freesurfer, which is an open-source neuroimaging toolkit for processing, analyzing, and visualizing human brain MR images. <https://surfer.nmr.mgh.harvard.edu/fswiki/FreeSurferSoftwareLicense>.

Fig R1. A comparison of our original and revised results.

After we obtained these results, we again consulted Dr. Gang Chen, who suggested that we implement a Bayesian approach in our analysis, as the conventional adjustment methods for multiple comparisons result in an overly conservative penalty for multiple testing (due to treating each brain region as entirely unrelated to the rest of the regions). Dr. Chen advised us to implement another approach that he and his co-authors suggest in Chen et al. (2020)². However, due to the complexity of our data structure and the large number of observations, it was not computationally feasible to run the Bayesian approach with the available resources. We note this as a limitation in the Supplementary Information:

“We also attempted to model our data using the Bayesian multilevel-modeling approach of Chen et al.². However, due to the complexity of our data structure and the large number of observations, it was not computationally feasible to deploy this Bayesian approach with the available resources.” (pg. 9)

We have updated Fig. 1 in the manuscript to replace the original results (see Fig. R1a) with the new results (see Fig. R1b). Additionally, we have changed our discussions of Study 1 results to focus on the regions with the largest effects in the new models. We have also updated Supplementary Table 2 and Supplementary Figs. 1 and 2 with results using the new approach.

Following Reviewer 1's suggestions, we also conducted a permutation-based approach, which shows a broadly similar pattern of results as Fig. R2 (see comment R1.3).

R1.3. The multiple comparison approach the paper takes in Study 1 is excessively conservative. At the very least, I would suggest using the Holm-Bonferroni procedure, instead of the regular Bonferroni correction – Holm-Bonferroni has the exact same assumptions as Bonferroni, but offers slightly more statistical power. However, the ideal solution would be to use a correction method that can benefit from the dependencies between tests (i.e., dependencies between brain regions in the parcellation, and between contrasts). Using maximal statistical permutation testing would be one such approach. This could be achieved here via a modified version of the Mantel test. Importantly, this would need to be modified to appropriately deal with the structure of the data. Rather than just a single permutation of rows/columns, there would need to be a permutation first of subjects, then of a second permutation of videos, in order to generate an appropriately structured null matrix. The mixed effects model would be fit again on each iteration of the permutation procedure, and the maximum t-statistic across regions/contrasts computed and added to the null distribution. In addition to being less conservative, an additional advantage of taking this approach is that it would obviate the need to correct degrees of freedom to obtain p-values, as discussed above.

Thank you for your suggestions. As suggested by Reviewer 1, we now apply the Holm-Bonferroni procedure instead of the regular Bonferroni correction to the revised results throughout the manuscript. See, for example, Fig. R1b on the preceding page of this letter. We also conducted a modified version of the Mantel test, as suggested by the reviewer. We permuted the data 5000 times to create a null distribution of the data while accounting for the dependency structure of our data due to repeating subjects and videos. Specifically, for each permutation, we first uniformly randomly shuffled the subject identifier while keeping the brain data constant. We then uniformly randomly shuffled the video identifier while keeping the brain data constant. We then ran the original mixed-effects model. For each permutation, we added the maximum t-statistic across regions and contrasts to a null distribution. We then compared the actual t-statistics from the un-permuted data to those from the null distribution. We show the results of this procedure in Fig. R2. The results from this permutation-based approach are qualitatively very similar to our findings using our original method and our newly-updated mixed-effects approach (see Fig R1), although fewer parcels emerged as significant using this approach.

We have added a short discussion of this alternative method and the findings to the Supplementary Information:

“To test the robustness of our results to alternative statistical-modeling approaches, we also conducted a modified version of the Mantel test of our primary analyses. First, we permuted the data 5000 times to create a null distribution of the data that accounts for the dependence structure of the data from repeating subjects and videos. Specifically, for each permutation, we first uniformly randomly shuffled the subject identifier while

Editorial Note: Figure R2 in this Peer Review File was created using Freesurfer, which is an open-source neuroimaging toolkit for processing, analyzing, and visualizing human brain MR images. <https://surfer.nmr.mgh.harvard.edu/fswiki/FreeSurferSoftwareLicense>.

keeping the brain data constant. We then uniformly randomly shuffled the video identifier while keeping the brain data constant. We then ran the original mixed-effects model that we reported in the main manuscript. For each permutation, we added the maximum t-statistic across regions and contrasts to a null distribution. We then compared the actual t-statistics from the un-permuted data to those from the null distribution. We show the results of this procedure in Supplementary Fig. 3. The results from this permutation-based approach are similar to our findings with the method that we reported in the main manuscript, although fewer parcels emerged as significant using this approach.” (pg. 9)

Fig R2 (also Supplementary Fig. 3). Relating neural similarity and sharing likelihood using a modified Mantel test with a maximum t-statistic approach. Using this alternative statistical-modeling approach, we obtained a similar pattern of results to the ones that we reported in the main manuscript.

R1.4. The paper would be improved by a limitations/constraints on generalizability paragraph in the discussion. It feels as though there are likely to be some important contexts in which the lesson learned here may not apply. For example, if I want to hear a critique of a piece of content, I might be more likely to share it with someone who I expect to have a different viewpoint than my own than with someone who shares my opinion. There are also common limitations of sample demographics and content topics which should nonetheless be acknowledged.

Thank you for this comment. To address these limitations, we have included the following paragraph in the Discussion section.

“It also remains unclear whether our results still apply in contexts in which individuals have overt motivations to seek different viewpoints from their own when sharing content (e.g., when one seeks critiques of content or is unsure of how to interpret content). In such contexts, perceived similarity in viewpoints with others may not be a key driver of information sharing. Furthermore, Study 1 participants were young adults, and Studies 2 and 3 used online convenience samples in the United States. Future work can clarify whether our findings generalize across diverse contexts and populations. Moreover, future studies that include other forms of information sharing that do not involve social media may provide further insight into whether our findings also hold for other sharing contexts (e.g., offline sharing of information by people who are not regular users of social media).” (pg. 19–20)

We have also expanded our discussion regarding the content topics to include controversial content in the following sentences:

“The stimuli in our studies included a variety of different topics and themes (e.g., a scientific demonstration, comedy clips, and social issues in Study 1; see Supplementary Table 1). Therefore, we are unable to make strong claims about specific message-level characteristics that may influence the effects that we found between perceived similarity and sharing likelihood. However, our results illustrate that content—regardless of its specific theme or domain—is more likely to be shared when individuals expect others to interpret the content similarly to themselves. We see this in the coordinated neural responses in Study 1, the self-report data in Study 2, and the experimental manipulation in Study 3. Our findings highlight fundamental neurobiological and psychological processes that motivate and predict sharing behavior across different content characteristics. Future work that explores these effects for different types of content (e.g., political content, morally-charged content, **controversial content**, and others) can further test whether the relationship between perceived similarity and sharing likelihood is affected by the content type (e.g., if these effects are heightened or reduced in certain contexts).” (pg. 18–19)

Comments from Reviewer 2

Study 1 concerns:

R2.1. The authors suggest that the results of Study 1 demonstrate that similar neural responses of individuals in a social community are associated with a greater likelihood of sharing content. They also note that the results in Study 1 have potential alternative explanations, such as participants' engagement with the videos. It makes sense to me that individuals would like to share a video they are engaged with, and previous studies suggest that when individuals are more engaged, the stimulus synchronized them more. I don't see how Study 2 rules out the role

of engagement Study 1 results. I still think that Study 1 results could be explained by engagement per-se.

Thank you for this comment. We agree with Reviewer 2 that the Study 1 results leave open the possibility that participants were oblivious to which videos their community members responded similarly to (which could be the case if engagement were driving both neural similarity and sharing likelihood). This is the reason that we conducted Study 2. We sought to directly test a link between *perceived* similarity and sharing intentions. In other words, we designed Study 2 to explicitly test one of the multiple explanations for our findings in Study 1 — namely, that people are more likely to share content that they perceive will be similarly interpreted by others in their social circles.

We have expanded our discussion of this possibility in the following sentences:

“Notably, the results in Study 1 have potential alternative explanations. For example, when an individual finds that particular content is engaging, there can be both less mind-wandering (and hence greater alignment with others’ neural responses³⁰) and a greater desire to share that content. This latter possibility does not require participants to be aware that the content that they rate as more worthy of sharing also elicits similar responses in others. **When content is engaging, people may both have especially similar neural responses to it and be particularly likely to share it with others without necessarily realizing that the content may evoke very similar responses across perceivers.** Therefore, in Study 2, we directly tested the hypothesis that people are more inclined to share content to which they believe that others in their social circles will have similar responses through a pre-registered online behavioral study of 100 participants.” (pg. 9)

Furthermore, due to concerns that we share with Reviewer 2 that content that is more engaging evokes greater neural similarity and also is more likely to be shared, in Study 2, we explicitly controlled for the extent to which individuals found the content to be interesting.

In our initial submission, we noted this in the following sentences:

“Given prior work that suggests links between information sharing and both the valence of content and the extent to which that content is perceived as interesting^{25–27}, we also fit a linear mixed-effects model with sharing likelihood as the dependent variable, perceived similarity ratings as the independent variable, and participants’ interest and valence ratings as control variables. We found that the association between perceived similarity and sharing likelihood remained significant even after controlling for interest and valence ratings ($\beta = 0.189$, $SE = 0.038$, $p < 0.001$). This suggests that the link between perceived similarity and sharing likelihood does not arise merely because people are more likely to share and to have similar perceptions of information that is more interesting, extremely positive, or extremely negative.” (pg. 10)

To further clarify this point, we have also added a sentence to the introduction:

“Accordingly, we conducted an online behavioral study (Study 2) and found that participants were especially likely to share content when they believed that other people in their social circles would have similar views about the content as themselves. **These results held even when controlling for participants’ levels of interest in the content and for their evaluations of it.**” (pg. 5)

R2.2. I am not sure why the contrast made in Study 1 is between high and low sharing pairs. It seems to me that in order to test their hypothesis, the contrast should be between individuals that are more or less similar to each other (regardless of sharing), which I assume will resemble the contrast done in the authors' previous paper on this dataset (Parkinson, C., Kleinbaum, A. M., & Wheatley, T. (2018). Similar neural responses predict friendship. Nature communications).

Thank you for the opportunity to clarify this point. The main purpose of Study 1 was to test whether or not individuals would be more likely to share content when the content evoked neural responses that were similar to each other. In other words, we were interested in whether content that is more likely to be shared elicits higher neural similarity *between* individuals while content that is less likely to be shared elicits lower neural similarity *between* individuals. We include this point in the introduction:

“In Study 1, we used functional magnetic resonance imaging (fMRI) to test whether or not people are more likely to share content when it evokes similar neural responses in members of their social circles.” (pg. 4)

Study 1 was not intended to test whether individuals who are similar to each other in some behavioral attribute (such as people who are friends, in the paper that the reviewer cites) would also have similar neural responses. Study 1 was also not intended to test whether individual differences in information sharing is linked with neural similarity (i.e., whether people who share a lot of information at baseline have greater similarity with each other than people who do not share as much).

Therefore, the contrast that we used was at the *level of video–dyad combinations*. For each *unique dyad*, we categorized *for each video* whether both individuals in the dyad had a high sharing likelihood ({high sharing, high sharing}), both individuals in the dyad had a low sharing likelihood ({low sharing, low sharing}), or if one participant had a high sharing likelihood and the other had a low sharing likelihood ({low sharing, high sharing}).

To make these distinctions clearer, we have added the following sentences:

“Unlike existing studies, which have investigated whether or not similarities in a participant-level attribute (e.g., their number of friends²¹ or loneliness²⁷) are linked with greater neural similarity, we are interested in whether or not people are more likely to share content when it evokes similar neural responses in individuals in their social circles. Therefore, our contrasts are at the level of video–dyad combinations.” (pg. 7)

We also explicitly note this point in the following places in the manuscript:

“For each video, we categorized a dyad’s sharing-likelihood rating as (1) {high sharing, high sharing} if both participants in the dyad had a high likelihood of sharing the video, (2) {low sharing, low sharing} if both participants in the dyad had a low likelihood of sharing the video, and (3) {low sharing, high sharing} if one participant had a high likelihood of sharing the video and the other had a low likelihood of sharing it.” (pg. 6–7)

“We then conducted a planned-contrast analysis²⁹ to identify brain regions for which a high sharing likelihood is associated with more coordinated neural responses than a low sharing likelihood (i.e., $ISC_{\{high\ sharing, high\ sharing}} > ISC_{\{low\ sharing, low\ sharing}})$). We focus on the contrast $ISC_{\{high\ sharing, high\ sharing}} > ISC_{\{low\ sharing, low\ sharing}}$, as this contrast is our most direct test of the hypothesis that people are more likely to share content that different individuals interpret similarly than to share content that different individuals do not interpret similarly.” (pg. 7)

We also want to clarify that the employed data set for this paper is a different data set than the one that Reviewer 2 noted in their comment (“which I assume will resemble the contrast done in the authors’ previous paper on this dataset (Parkinson, C., Kleinbaum, A. M., & Wheatley, T. (2018). Similar neural responses predict friendship. *Nature communications*)”). The prior data set did not include any measures of sharing intentions.

R2.3. Did the authors look at differences in the association between sharing a video and neural synchronization between the different videos? I think it could be interesting to test the hypothesis that videos that elicit more controversial interpretations would show increased such association.

As we note in the manuscript, participants saw videos that span multiple content topics, have different durations, and were presented in a fixed order. Accordingly, our study design does not allow us to test whether or not these effects were stronger for certain types of content than for other types. We agree that this would be an interesting future direction, and we also note that point in the manuscript. We have clarified this point in the following parts of the manuscript. We copy them below for your reference.

“In Study 1, the videos were not presented in isolation; instead, they were presented in a fixed order amidst a stream of other content.” (pg. 19)

“The stimuli in our studies included a variety of different topics and themes (e.g., a scientific demonstration, comedy clips, and social issues in Study 1; see Supplementary Table 1). Therefore, we are unable to make strong claims about specific message-level characteristics that may influence the effects that we found between perceived similarity and sharing likelihood. However, our results illustrate that content—regardless of its specific theme or domain—is more likely to be shared when individuals expect others to interpret the content similarly to themselves. We see this in the coordinated neural responses in Study 1, the self-report data in Study 2, and the experimental manipulation in Study 3. Our findings highlight fundamental

neurobiological and psychological processes that motivate and predict sharing behavior across different content characteristics. Future work that explores these effects for different types of content (e.g., political content, morally-charged content, **controversial content**, and others) can further test whether the relationship between perceived similarity and sharing likelihood is affected by the content type (e.g., if these effects are heightened or reduced in certain contexts).” (pg. 18–19)

We have also added the following sentences to the Discussion section in response both to this comment and to a related comment by Reviewer 1. These sentences discuss the potential generalizability of our findings to controversial content:

“It also remains unclear whether our results still apply in contexts in which individuals have overt motivations to seek different viewpoints from their own when sharing content (e.g., when one seeks critiques of content or is unsure of how to interpret content). In such contexts, perceived similarity in viewpoints with others may not be a key driver of information sharing. Furthermore, Study 1 participants were young adults, and Studies 2 and 3 used online convenience samples in the United States. Future work can clarify whether our findings generalize across diverse contexts and populations. Moreover, future studies that include other forms of information sharing that do not involve social media may provide further insight into whether our findings also hold for other sharing contexts (e.g., offline sharing of information by people who are not regular users of social media).” (pg. 19–20)

Below we include one table that shows the results that relate ISCs in the right temporoparietal junction parcel with the sharing likelihood for each video. We chose this parcel because ISCs in this brain region were consistently linked to sharing likelihood across our analyses. Given the limitations that we noted above (e.g., potential order effects), we have decided to not include the table in the manuscript itself. However, we would be happy to integrate these results into the Supplementary Information if you think it would be helpful.

Results that relate ISCs with the binarized sharing likelihood variable ($ISC_{\{high\ sharing, high\ sharing}\} > ISC_{\{low\ sharing, low\ sharing}\}$): Video-level results for right temporoparietal junction (rTPJ)

Video	B	SE	p (uncorrected)	p (Holm–Bonferroni corrected)
Video 1	0.315	0.255	0.004†	0.060†
Video 2	-0.142	0.176	0.253	>0.250
Video 3	0.232	0.224	0.143	>0.250
Video 4	-0.019	0.228	0.906	>0.250
Video 5	-0.350	0.343	0.149	>0.250

Video 6	0.065	0.154	0.547	>0.250
Video 7	-0.060	0.298	0.777	>0.250
Video 8	0.310	0.241	0.070†	>0.250
Video 9	0.203	0.274	0.296	>0.250
Video 10	0.200	0.300	0.346	>0.250
Video 11	0.144	0.281	0.469	>0.250
Video 12	0.127	0.279	0.519	>0.250
Video 13	0.110	0.316	0.316	>0.250
Video 14	-0.037	0.107	0.625	>0.250

† $p < 0.10$; see Supplementary Table 1 for descriptions of each stimuli.

R2.4. Participants provided their likelihood to share each video outside the scanner, approximately an hour after they saw the video, and after they saw all the 14 videos in the scanner. Do the authors have any way to verify that the "sharing" ratings were not affected by the comparisons between the 14 videos, and indeed reflect participant's likelihood to share the video immediately after watching it? I think this point is relevant because the order of the videos was always the same, and it could affect participant's likelihood to share the video in retrospect.

Thank you for this comment. We shared the concern that participants' likelihood to share a video might be affected by the order in which the videos were presented. Therefore, we conducted permutation tests to determine if there was a significant relationship between sharing likelihood and when a video appeared in the stimulus sequence. These permutation tests indicate that there was no relationship between sharing likelihood and video order. See "Supplementary methods 1 for analyses in Study 1" in the Supplementary Information.

It is possible that the comparisons of the 14 videos may have affected participants' likelihood to share, but we believe that this is akin to what happens in everyday life, where individuals watch many different videos on social-media platforms (e.g., TikTok, YouTube, and Instagram) and in practice still compare them to determine a video's shareworthiness. We have added the following paragraph to the Discussion section to address this point:

"In Study 1, the videos were not presented in isolation; instead, they were presented in a fixed order amidst a stream of other content. Therefore, comparisons of the 14 videos in Study 1 may have influenced participants' likelihood to share. Although this setting has analogues in daily life experiences, where individuals watch videos on a variety of social media (e.g., TikTok, YouTube, and Instagram) in sequences that are affected by platforms' algorithms and still compare pieces of content to one another when

determining shareworthiness, future work can help elucidate the effects of contextual factors (such as the order in which stimuli are presented) on sharing likelihood.” (pg. 19)

With that said, we agree that a limitation of Study 1 is that all participants saw the videos in the same order, as mentioned in our response to the previous comment (R2.3). Therefore, in Study 2, we randomized the stimulus order and participants answered questions about their likelihood to share each content shortly after seeing the stimulus. The findings of Study 2 align with our hypothesis that we generated based on the results of Study 1 — namely, participants are more likely to share content that they *perceive* will be interpreted similarly by others. We have clarified this point by adding the following sentences to the manuscript:

“To address limitations in Study 1 from the fixed order of the stimuli and the time gaps between stimulus presentations and sharing-likelihood ratings, we randomized the order of the stimuli in Study 2. Additionally, participants answered questions about their likelihood to share each piece of content shortly after viewing the stimuli. See the Methods section for more details.” (pg. 10)

R2.5. In their binarizing procedure, why did the authors decide to attribute "3" sharing-likelihood to "high likelihood" (Pg 24, Ln 484-5)? If I understood correctly, the scale was 1-5, thus the value "3" is relatively neutral in terms of sharing.

Thank you for the opportunity to clarify this point. The mean sharing-likelihood rating was 2.06 and the median was 2, thereby justifying our approach of attributing a sharing-likelihood rating of 3 as “high”. We have added this information to the Methods section:

“The mean sharing-likelihood rating was 2.06 and the median was 2, so we classified sharing-likelihood ratings of 1 and 2 as “low likelihood” and sharing-likelihood ratings of at least 3 as “high likelihood”.” (pg. 25)

With that said, we include in Fig. R3 our analysis with subset data that excludes sharing-likelihood ratings of 3. This subset data thus includes only ratings of 1, 2 (which were classified as “low likelihood”) and 4, 5 (which were classified as “high likelihood”). The results of this analysis using this subset data show patterns of results that are very similar to our reported results.

Editorial Note: Figure R3 in this Peer Review File was created using Freesurfer, which is an open-source neuroimaging toolkit for processing, analyzing, and visualizing human brain MR images. <https://surfer.nmr.mgh.harvard.edu/fswiki/FreeSurferSoftwareLicense>.

Fig R3. Results excluding sharing-likelihood ratings of 3 that we described in R2.5. The patterns of results that we obtained using subset data excluding sharing-likelihood ratings of 3 are similar to the ones that we reported in the main manuscript.

R2.6. How did the authors took into account the fact that there was a (large) difference in the unique pairs of ratings? There were 3,485 {high sharing, high sharing}, 14,963 {low sharing, low sharing}, and 11,193 {low sharing, high sharing}.

Thank you for this comment. To address this issue, we did the following analysis. We first undersampled the data uniformly at random from the {low sharing, low sharing} and {low sharing, high sharing} observations to match the number of observations for the {high sharing, high sharing} observations. We then fit our main model on this portion of the data set with matching observations across the levels of the sharing variable. We repeated this process 1000 times. We then averaged across the 1000 estimates for the contrast of interest (namely, {high sharing, high sharing} > {low sharing, low sharing}). We show the results in Fig. R4. As the figure indicates, the results of these analyses that use subset data of matching observations across different levels of sharing are very similar to our main results. We have added these results to the Supplementary Information.

Editorial Note: Figure R4 in this Peer Review File was created using Freesurfer, which is an open-source neuroimaging toolkit for processing, analyzing, and visualizing human brain MR images. <https://surfer.nmr.mgh.harvard.edu/fswiki/FreeSurferSoftwareLicense>.

Results with matching numbers of observations

Fig R4 (also Supplementary Fig. 4). Relating neural similarity and sharing likelihood with matching numbers of observations. The patterns of results that we obtained using subset data with matching observations of our dyad-level sharing-likelihood variable are similar to the ones that we reported in the main manuscript.

R2.7. I wonder why the authors used such a conservative threshold for multiple comparisons? Which p-value is considered significant in the correction they used (Bonferroni-corrected the p-values for multiple comparisons at $p < 0.001$)? Sometimes too conservative threshold can bias the results. For example, theoretically it could be that in some parcels the low-sharing elicited higher brain synchronization than the high-sharing, but the effect was not as significant.

Thank you for this feedback. In response to Reviewer 1's similar suggestions, we reran our analyses to expand our model to include additional terms and used the less conservative Holm-Bonferroni correction. The Holm-Bonferroni correction at $p = 0.05$ corresponds to approximately $p = 0.00023$ uncorrected. We show these new results in Fig. R1b in our response to comment R1.2 earlier in this document. As the figure indicates, there were no significant parcels in the cortical brain in which higher intersubject correlations were linked to lower sharing likelihoods.

R2.8. Did the authors measure how much the participants in this study use social media? I wonder if there are people that do not use social media (or rarely use it) and this would affect their ratings of how likely they are to share the piece.

Thank you for your comment regarding the baseline usage of social media by our study participants. We recognize that the extent to which participants use social media may influence their likelihood of sharing content. Of the 66 participants in our study, only two rated their likelihood to share all five articles as a "1". This indicates that the vast majority of our participants demonstrated variability in their sharing likelihoods. Furthermore, had participants indicated low levels of sharing-likelihood ratings, it would likely hinder our ability to detect effects due to the lack of variability in sharing ratings.

With that said, we agree that it would be interesting to test these relationships for other forms of sharing that do not involve social media. We have added the following sentences in the Discussion section to suggest such research as a future direction:

“Moreover, future studies that include other forms of information sharing that do not involve social media may provide further insight into whether our findings also hold for other sharing contexts (e.g., offline sharing of information by people who are not regular users of social media).” (pg. 19–20)

Study 2 concerns:

R2.9. I could not locate the topics of the five articles used as stimuli. This is critical, because it is vital to understand to what extent they were controversial, and in what manner. To take an extreme example, I would imagine that it would be much easier to share a National Geographic article about Panda Bears with someone who is not similar to me (or is this perhaps also controversial these days?) than a New York Times piece about Trump's trial.

Thank you for this feedback. We have clarified in our Methods section that each participant saw 5 news articles that we chose uniformly at random from a sample of 29 news articles that we had pretested in a pilot study. We selected the news articles to (1) possess a range in the extent to which their content would elicit similar interpretations across individuals and (2) be somewhat interesting, as articles that are widely perceived to be boring, irrelevant, or uninteresting are unlikely to be shared, even when interpretations of the content are very similar across individuals.

All news articles are from www.nytimes.com, www.npr.org, or www.wsj.com and were published within the 6 months of the time that we conducted our study. (We conducted Study 2 in January 2021.) We have placed hyperlinks to the articles that were used as stimuli for Study 2 in a repository (at the website <https://zenodo.org/records/11106495>), and we also now include this hyperlink in our manuscript. We had hoped to include the articles themselves, but we are not permitted to post the original articles online due to concerns about copyright infringement.

We have also modified the Methods section to clarify the procedure in Study 2:

“All participants saw the headlines and abstracts (i.e., short summaries) of five different news articles **that were chosen uniformly at random from a sample of 29 news articles that we pretested in a pilot study to ensure that they (1) ranged in the extent to which their content would elicit similarity in interpretations across individuals and (2) were somewhat interesting, given that articles that are widely perceived to be uninteresting are unlikely to be shared (as a baseline) irrespective of how one believes others will interpret it. Hyperlinks to the stimuli are available at <https://zenodo.org/records/13799055>.**” (pg. 27)

As Reviewer 2 notes, some of the news articles, such as those that are political in nature, may be perceived as more controversial than others. However, the purpose of Study 2 was to investigate whether content—regardless of the specific theme or domain of that content—is more likely to be shared when individuals expect others to interpret the content similarly to themselves. We agree with Reviewer 2 that this leaves an important gap for future studies to investigate how the relationship between perceived similarity and sharing likelihood may be influenced by the type of content. Accordingly, we have also included the following text in the Discussion section to acknowledge the limitations of our study and to suggest this as a potential avenue for future work:

“Future work that explores these effects for different types of content (e.g., political content, morally-charged content, **controversial content**, and others) can further test whether the relationship between perceived similarity and sharing likelihood is affected by the content type (e.g., if these effects are heightened or reduced in certain contexts).

In Study 1, the videos were not presented in isolation; instead, they were presented in a fixed order amidst a stream of other content. Therefore, comparisons of the 14 videos in Study 1 may have influenced participants’ likelihood to share. Although this setting has analogues in daily life experiences, where individuals watch videos on a variety of social media (e.g., TikTok, YouTube, and Instagram) in sequences that are affected by platforms’ algorithms and still compare pieces of content to one another when determining shareworthiness, future work can help elucidate the effects of contextual factors (such as the order in which stimuli are presented) on sharing likelihood.

It also remains unclear whether our results still apply in contexts in which individuals have overt motivations to seek different viewpoints from their own when sharing content (e.g., when one seeks critiques of content or is unsure of how to interpret content). In such contexts, perceived similarity in viewpoints with others may not be a key driver of information sharing. Furthermore, Study 1 participants were young adults, and Studies 2 and 3 used online convenience samples in the United States. Future work can clarify whether our findings generalize across diverse contexts and populations. Moreover, future studies that include other forms of information sharing that do not involve social media may provide further insight into whether our findings also hold for other sharing contexts (e.g., offline sharing of information by people who are not regular users of social media).” (pg. 19–20)

R2.10. In Study 2, were the two questions asked immediately one after the other? ("Consider the people with whom you are friends with on social media. How confident are you that they would all generally share your views on the content of the article?" and the question " How likely would you be to share this article on social media (e.g., on your Facebook timeline, Instagram, or Twitter)?").

Moreover, why was the scale for these questions was 1-5, and not a VAS scale?
I suggest that these two characteristics of the task made it easier for the participants to make similar ratings on both questions.

The two questions were asked immediately one after the other, although the order that the questions were presented to participants was selected uniformly at random (i.e., some participants first saw the sharing-likelihood question and then saw the perceived-similarity question, and other participants first saw the perceived-similarity question and then saw the sharing-likelihood question). The sharing-likelihood question was on a 1–5 Likert scale, whereas the perceived-similarity question was on a 1–100 scale. Therefore, we expect that it is very unlikely that the participants would rate the two items on the same scale.

With that said, we agree with Reviewer 2 that this is a potential issue. Future studies can have a distraction task in the middle of these two questions to alleviate concerns about the potential effects of posing the questions one after another. In our case, we experimentally manipulated perceived similarity in Study 3, rather than asking participants to rate perceived similarity immediately before or after asking them to rate their sharing likelihood. We believe that this mitigates the concern about consecutive item ratings facilitating similar responses across items and provides a robust test of our hypothesized relationships. We have added the following text to the revised manuscript to address these points:

"It is possible that asking participants about sharing likelihood and perceived similarity in close succession in Study 2 increased the chance that participants gave similar responses to both questions. Study 3 alleviates this concern by experimentally manipulating perceived similarity and having participants report only their sharing likelihoods." (pg. 12)

R2.11. As concern (8), did the authors measure how much the participants in this study use social media? I wonder if there are people that do not use social media (or rarely use it) and this would affect their ratings of how likely they are to share the piece.

For Study 2, to be eligible to participate, participants were required to have an account on social media and to report that they sometimes share information on social media. Specifically, they were required to answer "yes" to the following questions to be eligible to participate:

"Do you currently have an account on any of the following social media platforms: Facebook, Twitter, Instagram?"

“Do you agree with the following statement? I sometimes share news stories on social media (for example, on Facebook, Twitter, and/or Instagram).”

It was an oversight on our part to not state this explicitly in the manuscript. We appreciate Reviewer 2’s comment for the opportunity to clarify and add this important detail to our manuscript. We have added the following text to the manuscript:

“Participants were required to have an account on social media and to report that they sometimes share content (in this case, news stories) on social media. Specifically, to be eligible to participate, participants had to answer “yes” to both of the following questions: (1) “Do you currently have an account on any of the following social media platforms: Facebook, Twitter, Instagram?”; and (2) “Do you agree with the following statement? I sometimes share news stories on social media (for example, on Facebook, Twitter, and/or Instagram).”” (pg. 26–27)

As we noted in our response to R2.8, we also edited our manuscript to include the following text:

“Moreover, future studies that include other forms of information sharing that do not involve social media may provide further insight into whether our findings also hold for other sharing contexts (e.g., offline sharing of information by people who are not regular users of social media).” (pg. 19–20)

Study 3 Concerns

R2.12. Again, as in concern (9), I could not locate the topics of news articles. This could affect the interpretation of Study 3 results.

Thank you for this feedback. As in Study 2, we chose the five news articles that participants saw to (1) have a range in the extent to which the content would elicit similar interpretations across individuals and (2) be somewhat interesting, given that articles that are widely perceived to be uninteresting are unlikely to be shared in the first place. The articles were all from www.nytimes.com and were published within 9 months of the time that we conducted the study. (We conducted Study 3 in June 2022.)

We now include hyperlinks to the articles that were used as stimuli for Study 3 at the website <https://zenodo.org/records/11432271>.

We have also modified the text in the Methods section to clarify our procedure for Study 3:

“As in Study 2, the five news articles were chosen to (1) range in the extent to which the content would elicit similarity in interpretations across individuals and (2) be somewhat interesting, given that articles that are widely perceived to be uninteresting are unlikely to be shared (as a baseline) irrespective of with whom one is considering sharing such

articles. **Hyperlinks to the stimuli are available at <https://zenodo.org/records/13799122>.**“ (pg. 29–30)

As we mentioned in our response to R2.9, we also note the limitations of our stimuli in the following text in the Discussion section:

“Future work that explores these effects for different types of content (e.g., political content, morally-charged content, **controversial content**, and others) can further test whether the relationship between perceived similarity and sharing likelihood is affected by the content type (e.g., if these effects are heightened or reduced in certain contexts).” (pg. 19)

“It also remains unclear whether our results still apply in contexts in which individuals have overt motivations to seek different viewpoints from their own when sharing content (e.g., when one seeks critiques of content or is unsure of how to interpret content). In such contexts, perceived similarity in viewpoints with others may not be a key driver of information sharing. Furthermore, Study 1 participants were young adults, and Studies 2 and 3 used online convenience samples in the United States. Future work can clarify whether our findings generalize across diverse contexts and populations. Moreover, future studies that include other forms of information sharing that do not involve social media may provide further insight into whether our findings also hold for other sharing contexts (e.g., offline sharing of information by people who are not regular users of social media).” (pg. 19–20)

R2.13. Both Study 2 and Study 3 reveal that individuals were more likely to share information when they believed that others in their social circles would hold similar viewpoints as themselves about the information. I think this is an interesting finding, that is very much in line with previous literature about the need to belong. However, I am not sure if the studies' results highlight a new perspective on it.

Thank you for this opportunity to clarify our contribution to this literature. While it is true that our findings align with existing literature on the need to belong, our studies add new insights by explicitly quantifying the influence of perceived similarity on sharing behavior. In doing so, our findings can inform specific potential applications for studying various consequential phenomena in information sharing in ways beyond just general theorizing about the need to belong. For instance, our findings suggest that perceived similarity may drive the spread of misinformation, which can inform targeted interventions to prevent the spread of misinformation, as we discuss in the Discussion section. Consequently, our findings provide empirical evidence that extends the “need to belong” theory by highlighting the role of perceived similarity, thereby offering a more nuanced perspective on the social motivations that influence information sharing.

There are also many other perspectives and accounts that pose different theories that are not inherently social in nature about what motivates individuals to share information. For instance, one prominent non-social account emphasizes that people share information simply to spread

important information that fills a need for accuracy³⁻⁷. Therefore, our findings also contribute to the general theorizing about the motivations behind information sharing. Our findings emphasize social accounts in such theorizing.

We have added a few sentences to our Introduction and Discussion sections to emphasize our study's unique contributions. We would be happy to include additional text if Reviewer 2 thinks that it would be useful.

“In the present paper, we investigate the idea that motivations to achieve and maintain shared reality with others may play a critical role in information sharing. **We thereby provide empirical evidence that advances existing theories about the motivations behind information sharing, which have often focused on non-social drivers of sharing (e.g., the desire to spread information that fulfills a need for accuracy¹⁶⁻²⁰). Across three studies, we test the hypothesis that people are more likely to share information when they believe that others in their social circles will share their viewpoints and opinions about the information than when they believe that others' viewpoints will differ from theirs.**” (pg. 4)

“Accordingly, the results from our three studies corroborate theories of information sharing as an inherently social behavior^{5,13,45} that supports fundamental human motivations to connect and belong³, **rather than theories that emphasize non-social motivations (such as a desire for accuracy)¹⁶⁻²⁰.**” (pg. 18)

References

1. Chen, G., Taylor, P. A., Shin, Y. W., Reynolds, R. C. & Cox, R. W. Untangling the relatedness among correlations, Part II: Inter-subject correlation group analysis through linear mixed-effects modeling. *NeuroImage* **147**, 825–840 (2017).
2. Chen, G. *et al.* Untangling the relatedness among correlations, part III: Inter-subject correlation analysis through Bayesian multilevel modeling for naturalistic scanning. *NeuroImage* **216**, 116474 (2020).
3. Paletz, S. B. F., Auxier, B. E. & Golonka, E. M. *A Multidisciplinary Framework of Information Propagation Online*. (Springer International Publishing, Cham, Switzerland, 2019).
4. Berger, J. Word of mouth and interpersonal communication: A review and directions for future research. *Journal of Consumer Psychology* **24**, 586–607 (2014).
5. Lee, C. S. & Ma, L. News sharing in social media: The effect of gratifications and prior experience. *Computers in Human Behavior* **28**, 331–339 (2012).
6. Kim, D. H., Jones-Jang, S. M. & Kenski, K. Why do people share political information on social media? *Digital Journalism* **9**, 1123–1140 (2021).
7. Wong, L. Y. C. & Burkell, J. Motivations for Sharing News on Social Media. in *Proceedings of the 8th International Conference on Social Media & Society* 1–5 (ACM Press, Toronto, ON, Canada, 2017).